# Ume6 protein complexes connect morphogenesis, adherence and hypoxic genes to shape *Candida albicans* biofilm architecture

Eunsoo Do [1], C. Joel McManus [2], Robert Zarnowski [3], Manning Y. Huang [2,4], Katharina Goerlich[1], David R. Andes [3] & Aaron P. Mitchell [1] ✉

Biofilms of the fungal pathogen *Candida albicans* can form on implanted medical devices and contribute to fungal virulence and are recalcitrant to antifungal therapy. The transcription factor Ume6 directs hyphal cell elongation and thus promotes biofilm formation in *C. albicans*. However, how exactly this key biofilm and virulence regulator functions has remained unclear. Here RNA sequencing and chromatin immunoprecipitation with sequencing data show that Ume6 binds to and activates multiple biofilm-relevant genes. Ume6-associated sequence motifs correspond to binding sites for biofilm master regulators Efg1 and Ndt80, and hypoxic response regulator Upc2. Co-immunoprecipitation assays show the existence of Ume6–Efg1, Ume6–Ndt80 and Ume6–Upc2 protein complexes. Promoter binding of Ume6 is partially dependent upon Efg1, Ndt80 or Upc2, as is Ume6 target gene activation, thus indicating that the protein complexes function to drive Ume6–target gene interaction. Ume6 therefore acts as a bridge that connects the hyphal morphogenesis and adherence genes that shape biofilm architecture and the hypoxic response genes required for growth in the low-oxygen biofilm environment. These findings are vital for our understanding of the pathobiology of *C. albicans* and could open the way to new treatment options.

*Candida albicans* is a fungal commensal and pathogen of humans[1]. It lies in the World Health Organization critical priority group because of the high frequency and mortality of invasive candidiasis. A major risk factor for invasive candidiasis is the presence of an implanted medical device, which can serve as a substrate for biofilm formation[2,3]. Biofilms release cells that infect deep tissue, and are recalcitrant to antifungal therapy.

Biofilm formation by *C. albicans* depends upon its ability to produce filamentous hyphal cells[3,4]. Their properties arise from a set of

genes, called hypha-associated genes, that are expressed at much higher levels in hyphal cells than in non-filamentous yeast cells[5]. Hypha-associated genes contribute to biofilm formation and virulence through roles in adherence, biofilm extracellular matrix biogenesis, polarized growth and other processes[4]. An interconnected network of transcription factors (TFs) that activate hypha-associated genes, called the biofilm master regulators, are required for biofilm formation and filamentation[4]. Most of the master regulators have prion-like

[1]Department of Microbiology, University of Georgia, Athens, GA, USA. [2]Department of Biological Sciences, Carnegie Mellon University, Pittsburgh, PA, USA. [3]Department of Medical Microbiology and Immunology, University of Wisconsin, Madison, WI, USA. [4]Present address: Department of Biochemistry and Biophysics, University of California San Francisco, San Francisco, CA, USA. ✉e-mail: Aaron.Mitchell@uga.edu

domains (PrLDs), and several have been shown recently to assemble into phase-separated condensates in vitro and in heterologous hosts[6]. These findings, combined with elegant mutational analysis[6], indicate that master regulator complexes, either in the form of stable condensates or transient hubs, are critical for cells to undergo the transition from yeast to hyphae and produce biofilm.

One key target of the biofilm master regulators is the hypha-associated gene *UME6*, which specifies a zinc-cluster TF. Ume6 has a functional role distinct from that of the master regulators: it is required to extend hyphae but not to initiate hypha formation[7]. Yet it also has a property considered a defining feature of master regulators[8] in all organisms: engineered overexpression of *UME6* bypasses both environmental and genetic signals that otherwise control filamentation and biofilm formation[9,10]. In fact, *UME6* ortholog overexpression drives filamentous growth in every *Candida* pathogen tested[11,12]. The central role of Ume6 in host interaction is indicated by its requirement for virulence[7] and for priming systemic immunity in animal models[13]. Perhaps because of its powerful impact, *UME6* is tightly regulated at the levels of transcription[7], translation[14] and protein stability[15]. Despite its critical role in *C. albicans* pathogenic processes, there is limited information about the Ume6 mechanism of action. Elucidation of its mechanism and targets is vital for the understanding of *Candida* pathobiology. Here we report that Ume6 acts in protein complexes with two known biofilm regulators, Efg1 and Ndt80, as well as the hypoxic response regulator Upc2. Ume6 promoter binding and gene activation are stimulated by Efg1, Ndt80 and Upc2, facilitating expression of both hyphal genes and hypoxic response genes. Therefore, Ume6 bridges the hyphal morphogenesis and adherence genes that shape biofilm architecture, and the hypoxic response genes required for growth in the low-oxygen biofilm environment.

## Results

### Ume6 expression targets

The gene expression impact of *ume6Δ/Δ* mutations is variable among clinical *C. albicans* isolates[16]. Therefore, we used *UME6* overexpression in an *efg1Δ/Δ* master regulator mutant to define Ume6 transcriptional targets. The *efg1Δ/Δ* mutant is defective in filamentation, biofilm formation and expression of hypha-associated genes, including *UME6* (ref. 16). Previous studies showed that *UME6* overexpression in *efg1Δ/Δ* mutants can drive filamentation and biofilm formation, but not hypha-associated gene expression, under the conditions tested[9,17]. Here we conducted RNA sequencing (RNA-seq) analysis with strains carrying functional *UME6-HA* epitope-tagged alleles grown in strong inducing conditions for hyphae and biofilm (Roswell Park Memorial Institute medium (RPMI) + foetal bovine serum (FBS), 37 °C), using 4-h planktonic hyphal growth to represent early stages of biofilm formation. All told, we compared four strains: a wild type (*EFG1/EFG1 UME6-HA/UME6-HA*), a wild type with overexpressed *UME6* (*EFG1/EFG1 RBT5-UME6-HA/RBT5-UME6-HA*), an *efg1Δ/Δ* mutant (*efg1Δ/Δ UME6-HA/UME6-HA*) and an *efg1Δ/Δ* mutant with overexpressed *UME6* (*efg1Δ/Δ RBT5-UME6-HA/RBT5-UME6-HA*). The *RBT5* promoter, which is active in RPMI + FBS medium, was used for overexpression[9,17]. Western blot analysis confirmed that Ume6-haemagglutinin (HA) levels were undetectable in the *efg1Δ/Δ* mutant and elevated in the *efg1Δ/Δ* mutant with overexpressed *UME6* (Fig. 1a). Strains were validated with filamentation and biofilm formation assays (Fig. 1b). As expected, *UME6* overexpression restored filamentation and biofilm formation in the mutant *efg1Δ/Δ* context, and did not alter either property in the wild-type *EFG1/EFG1* context under these inducing conditions.

RNA-seq analysis (see 'Tab 1' in Supplementary Table 1) shows the expected severe downregulation of numerous hypha-associated genes (see 'Tab 5' in Supplementary Table 1) in the *efg1Δ/Δ* mutant compared with the wild type (Fig. 1c) and increased expression of these same genes in the *efg1Δ/Δ* mutant overexpressing *UME6* compared with the *efg1Δ/Δ* mutant (Fig. 1c). However, most hypha-associated gene RNA

levels were lower in the *efg1Δ/Δ* mutant overexpressing *UME6* than in the wild type or in the wild type overexpressing *UME6* (Fig. 1c and 'Tab 5' in Supplementary Table 1). This outcome indicates that Ume6 activates hypha-associated genes more efficiently in the presence of Efg1 than in its absence.

To define Ume6 direct target genes, we conducted chromatin immunoprecipitation with sequencing (ChIP-seq) analysis under these same growth conditions (see 'Tab 1' in Supplementary Table 2). Ume6-bound sites were defined by comparing the results with wild-type tagged (*EFG1/EFG1 UME6-HA/UME6-HA*) and untagged (*EFG1/EFG1 UME6/UME6*) strains in the SC5314 strain background. Binding peak locations correlated well with data from a wild-type tagged strain in the P75010 strain background (Fig. 2a and 'Tab 2' and 'Tab 9' in Supplementary Table 2) and from the *efg1Δ/Δ* mutant with overexpressed *UME6* (*efg1Δ/Δ RBT5-UME6-HA/RBT5-UME6-HA*) in the SC5314 background (Fig. 2b and 'Tab 5' and 'Tab 9' in Supplementary Table 2). This ChIP-seq comparison identified 1,206 bound genes for Ume6 (see 'Tab 1' and 'Tab 8' in Supplementary Table 2). Ume6-bound genes were enriched for Gene Ontology (GO) functional categories related to biofilm formation, filamentous growth and cell aggregation, and included 94 hypha-associated genes such as *ALS3*, *ECE1*, *HGC1*, *HWP1* and *HYR1*. In addition, Ume6 bound to promoter regions of biofilm master regulator genes *BCR1*, *BRG1*, *EFG1*, *FLO8* and *TEC1*, an indication that Ume6 is well integrated into the biofilm regulatory network. Ume6-bound genes also included ergosterol biosynthesis genes (*ERG1*, *ERG3*, *ERG4*, *ERG5*, *ERG6*, *ERG7*, *ERG11*, *ERG13*, *ERG2*, *ERG25*, *ERG251* and *HMG1*), a suggestion that Ume6 may have a broader functional role than previously known.

Ume6-bound genes significantly overlapped with previously identified Efg1-bound genes (Fig. 2c)[18]. Moreover, Ume6-bound genes included almost all Efg1 core targets, the genes bound and regulated by Efg1 across five diverse clinical isolates (Fig. 2d). This relationship between Ume6- and Efg1-bound genes suggests that Ume6 and Efg1 may interact in some way to control filamentation and biofilm formation.

### Efg1 dependence on Ume6 binding

We examined Ume6-bound sites in the four strains used for RNA-seq analysis to understand the impact of Efg1 on Ume6 binding. Genome-wide analysis showed weak Ume6 binding in the *efg1Δ/Δ* mutant owing to low *UME6* expression[16,18,19], and no overall difference in Ume6 binding events among the wild type, wild type overexpressing *UME6* and *efg1Δ/Δ* mutant overexpressing *UME6* (Fig. 2b). Average Ume6 binding intensities of the wild type overexpressing Ume6 and *efg1Δ/Δ* mutant overexpressing Ume6 were almost identical (Fig. 2b). Over most of the genome, Ume6 binds equally efficiently in the presence or absence of Efg1.

One subset of genes had distinct Ume6 binding properties. For 62 out of the 1,206 genes, expression levels and Ume6 binding were both significantly greater in the wild-type strain overexpressing *UME6* than in the *efg1Δ/Δ* strain overexpressing *UME6* (Fig. 2e and 'Tab 7' in Supplementary Table 2). This gene set was enriched for the GO terms related to biofilm formation. Binding profiles for *ECE1*, *HWP1*, *HGC1* and *HYR1* illustrate the promoter binding intensity differences (Fig. 2f). These results indicate that Ume6 binding to some gene promoters, including those of many hypha-associated genes, is more efficient in the presence of Efg1 than in its absence.

### Association of Ume6 with diverse binding motifs

Efg1 may improve Ume6 binding at some promoters through the formation of a protein complex. In fact, binding peaks of Ume6 and Efg1 coincide at numerous genes required for hypha and biofilm formation, including 83 of 110 Efg1 core targets (see 'Tab 8' in Supplementary Table 2). Figure 3a illustrates binding peak overlap for 12 genes tied functionally to biofilm and hypha formation. These data are consistent with the idea that Ume6 and Efg1 interact at a subset of target genes.

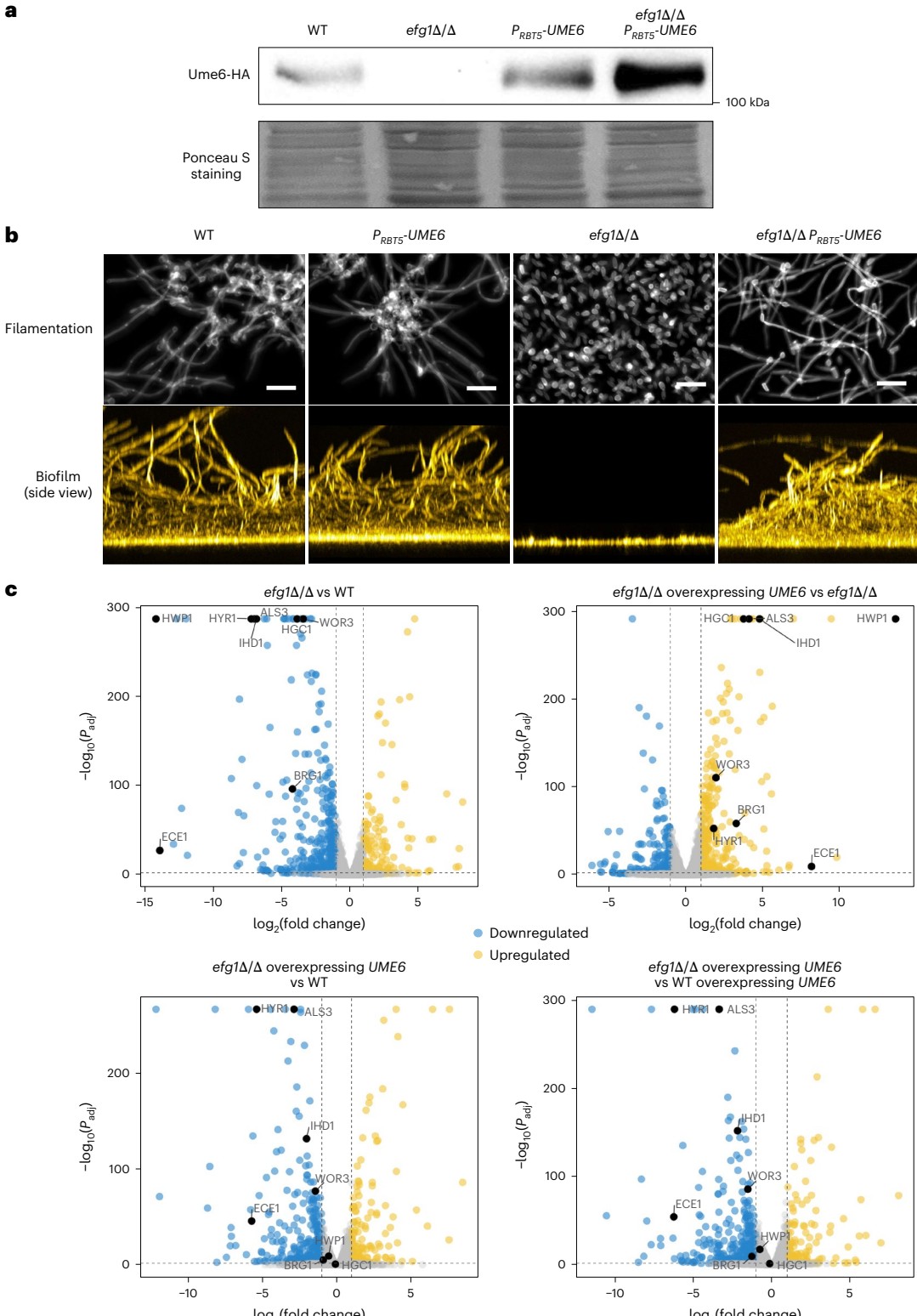

**Fig. 1 | Activation of hypha-associated genes is dependent on the presence of Efg1. a**, Western blot analysis was performed using cells grown in RPMI + FBS at 37 °C for 4 h. Ponceau S staining was used as a control representing the same amount of loaded protein. The images represent three independent experiments. **b**, For filamentation assay, cells were grown in RPMI + 10% FBS at 37 °C for 4 h and stained with calcofluor-white. Scale bars, 20 μm. For biofilm formation, cells were grown in RPMI + 10% FBS at 37 °C for 24 h and stained with calcofluor-white. Side-view projection images are shown. All images represent three independent experiments. **c**, Volcano plots showing differentially expressed genes in comparisons between two strains indicated at the top of each plot. The blue and yellow dots indicate genes that are significantly down- or upregulated, respectively (log$_2$(fold change) of greater than 1 or less than −1, adjusted P value ($P_{adj}$) < 0.05). Statistical significance was determined using two-sided Wald test, followed by multiple test correction using the Benjamini–Hochberg method for $P_{adj}$ in the DESeq2 package. All strains used in this figure are homozygous for alleles indicated. The wild-type (WT) genotype is *UME6-HA/UME6-HA*.

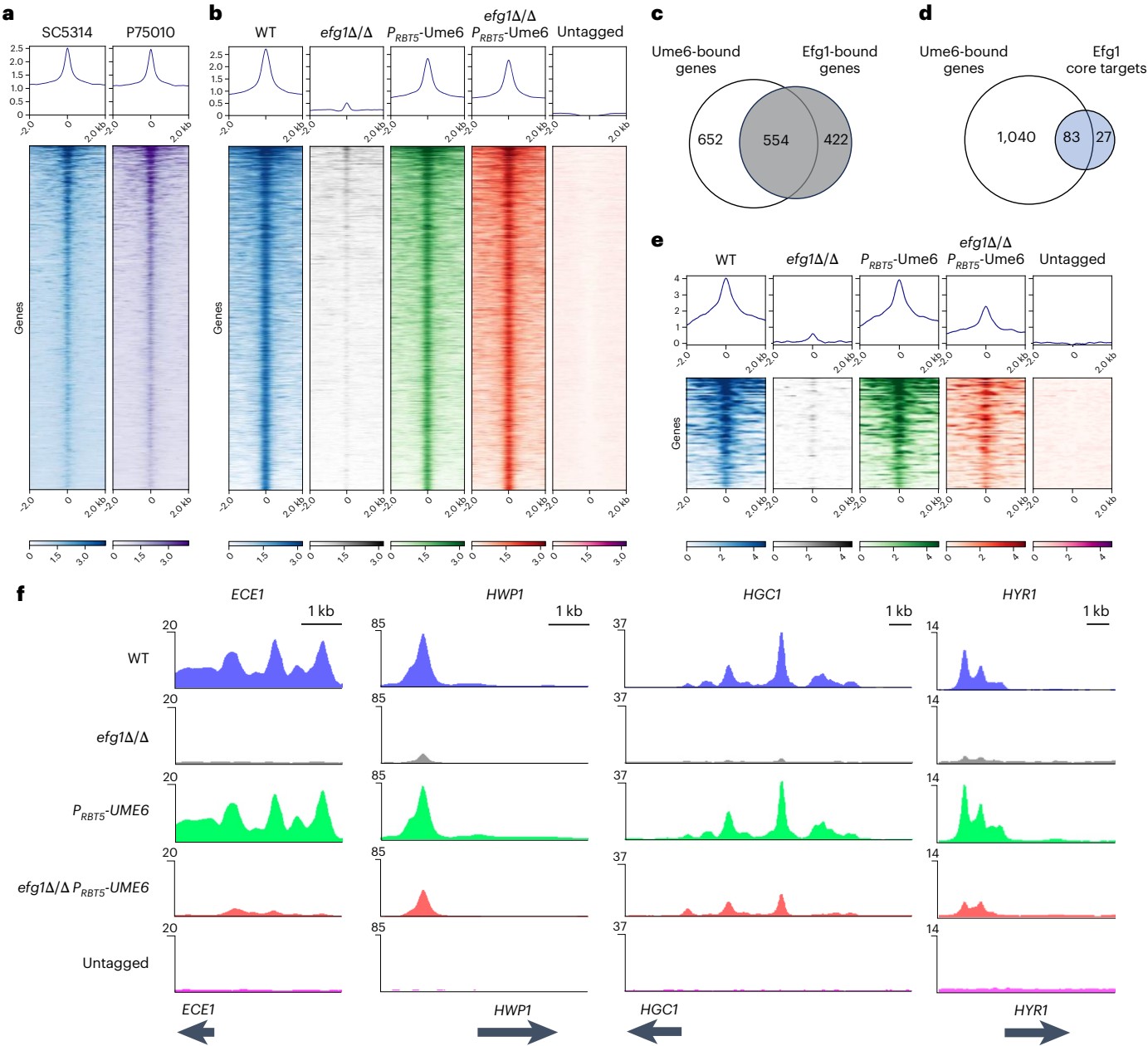

**Fig. 2 | Dependence of Ume6 binding on Efg1. a**, Heatmaps of the ChIP-seq data showing Ume6 occupancies in SC5314 and P75010 ranked by their binding affinity. Regions ±2 kb around Ume6 peak summits are shown. The average read coverage profiles are shown above the heatmaps. Pearson coefficients between SC5314 and P75010 are 0.83–0.87. Each heatmap represents three biological replicates ($n = 3$). **b**, Heatmaps of the ChIP-seq data showing Ume6 occupancies in the wild type, $efg1\Delta/\Delta$ mutant, wild type overexpressing *UME6* and $efg1\Delta/\Delta$ mutant overexpressing *UME6* in the genome. Regions ±2 kb around Ume6 peak summits are shown. The average read coverage profiles are shown above the heatmaps. Pearson coefficients between the wild type and $efg1\Delta/\Delta$ mutant with overexpressed *UME6* strains are 0.71–0.82. Each heatmap represents three biological replicates ($n = 3$). **c,d**, Venn diagrams depicting the intersection between Ume6-bound genes and either Efg1-bound genes or Efg1 core targets. The grey and blue circles indicate Efg1-bound genes and Efg1 core targets, respectively. Statistical significance was determined using Fisher's exact test

(significance for Ume6-bound genes and Efg1-bound genes: $P < 2.2 \times 10^{-16}$; significance for Ume6-bound genes and Efg1 core targets: $P < 2.2 \times 10^{-16}$). **e**, Heatmaps of the ChIP-seq data showing Ume6 occupancies at 62 targets in the wild type, $efg1\Delta/\Delta$ mutant, wild type overexpressing *UME6* and $efg1\Delta/\Delta$ mutant overexpressing *UME6* in the genome. Regions ±2 kb around Ume6 peak summits are shown. The average read coverage profiles are shown above the heatmaps. Each heatmap represents three biological replicates ($n = 3$). **f**, Genome browser tracks showing the Ume6 binding sites in the upstream region of hypha-associated genes *HWP1*, *HGC1*, *ALS3* and *ECE1* in each strain. The y axis indicates read counts. The track of the untagged strain was used as a control. The ChIP-seq tracks represent three biological replicates ($n = 3$). All strains used in this figure are homozygous for alleles indicated. The WT genotype is *UME6-HA/UME6-HA*. In **a**, **b** and **e**, each colour scale bar indicates the peak intensity, which is calculated as the log2 ratio of signals from the immunoprecipitated sample versus the input sample.

To explore Ume6 binding motifs, we first analysed sequences of all 1,681 Ume6 binding peaks. This analysis yielded many non-specific sequences, based on their high background levels or coverage of only a small fraction of bound regions. We then restricted our analysis to

a subset of the 429 strongest binding peaks (Methods). We detected three different motifs: 5′-ACACAAA-3′, 5′-TCGTCT-3′ and 5′-TGCAT-3′ (Fig. 3b), none of which resemble the *Saccharomyces cerevisiae* Ume6 binding motif 5′-GGCGGC-3′ (ref. 20). These motifs correspond to

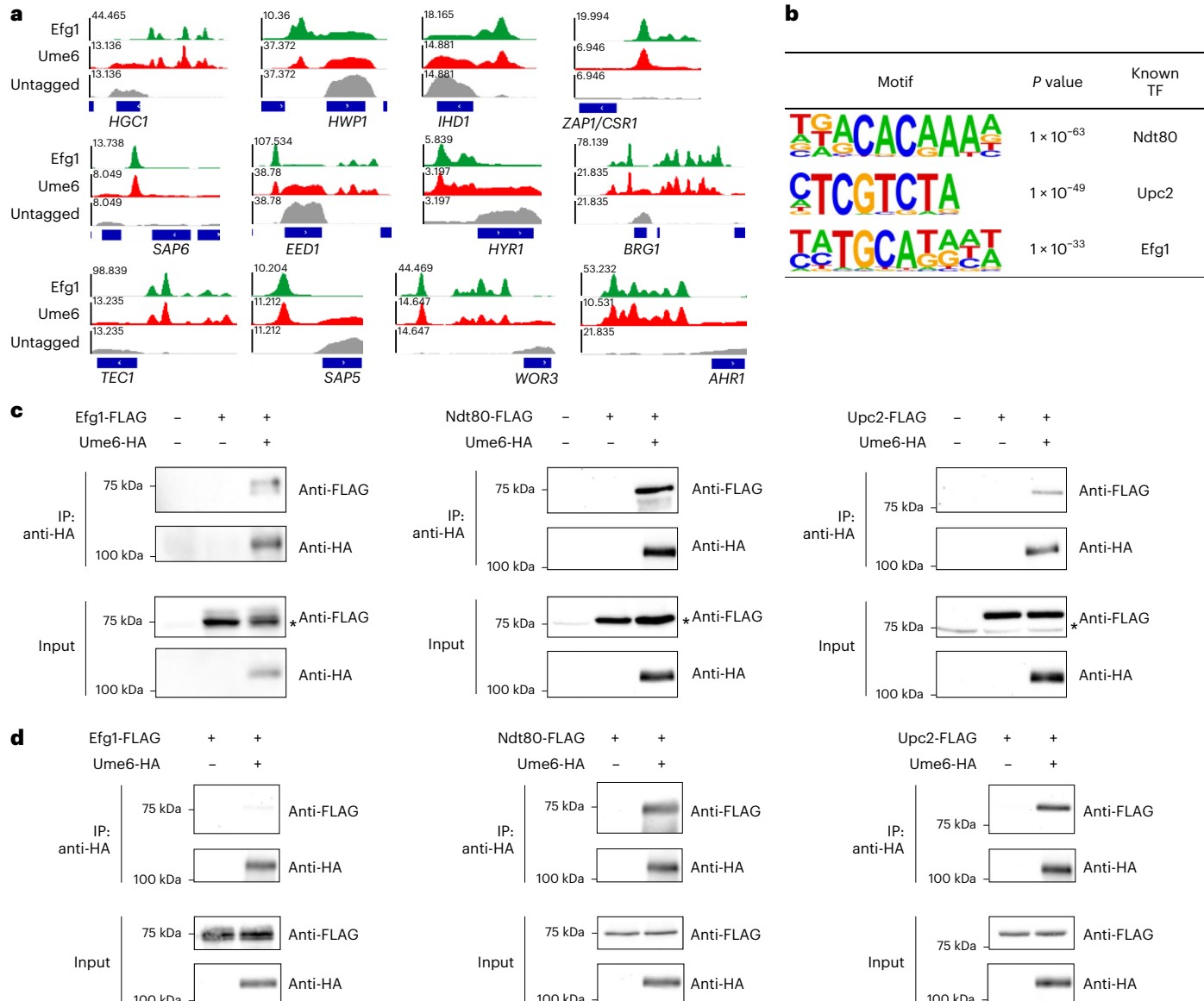

**Fig. 3 | Interaction of Ume6 with partner TFs Efg1, Ndt80 and Upc2. a**, Genome browser tracks for ChIP-seq data show the Efg1 and Ume6 binding regions in the upstream region of hypha-associated genes. The *y* axis indicates read counts. The track of the untagged strain was used as a control. The ChIP-seq tracks represent three biological replicates (*n* = 3). **b**, Binding motif analysis for SC5314 Ume6 was performed with HOMER. These three binding sites are corresponding to Ndt80, Upc2 and Efg1, respectively. **c,d**, Cells were grown in RPMI + 10% FBS at 37 °C for 4 h with shaking under planktonic conditions (**c**) or

for 24 h without shaking under biofilm conditions (**d**), and protein lysates from strains expressing the indicated HA- and/or FLAG-tagged protein were subjected to immunoprecipitation with anti-HA antibody. The precipitated proteins were detected using either anti-HA or anti-FLAG for western blot analysis. The cell lysates used for co-IP were used as an input control. Asterisks indicate non-specific bands. The images represent three independent experiments (*n* = 3). All strains used in this figure are homozygous for alleles indicated. The genotype of the untagged strain is *UME6*[+/+].

binding sites for biofilm master regulator Ndt80, hypoxia and ergosterol regulator Upc2, and Efg1, respectively[18,21,22]. Ume6-bound genes with Efg1 or Ndt80 binding motifs (Supplementary Table 3) were significantly enriched for GO terms related to hypha and biofilm formation. The Upc2 motif was found at fewer Ume6-bound genes than the other motifs, yet it was nonetheless associated with biofilm-related genes such as *ALS3*, *HWP1*, *HYR1*, *RHR2* and *ZAP1/CSR1* (Supplementary Table 3). Overall, these results suggest that DNA binding of Ume6 occurs in close proximity to Efg1, Ndt80 and Upc2.

## Ume6 interaction with Efg1, Ndt80 and Upc2

To investigate whether Ume6 physically interacts with Efg1, Ndt80 or Upc2, we conducted a co-immunoprecipitation (co-IP) assay. Anti-HA

antibody was used to precipitate tagged Ume6-HA, and anti-Flag antibody detected the presence of each Flag-tagged partner TF (Efg1, Ndt80 and Upc2). These experiments revealed the presence of three heterocomplexes: Ume6–Efg1, Ume6–Ndt80 and Ume6–Upc2 under planktonic hypha-inducing conditions (Fig. 3c) and under biofilm conditions (Fig. 3d). Recovery of Efg1 from a Ume6–Efg1 complex was stronger under planktonic conditions than under biofilm conditions. We were unable to detect a complex between Ume6 and the well-characterized iron regulator Sef1 (Extended Data Fig. 1), an indication that Ume6 is not simply a promiscuous interactor. These findings indicate that Ume6 interacts with Efg1, Ndt80 and Upc2. A simple hypothesis is that Ume6 recruitment to DNA is facilitated by these interactions.

## Ume6 C-terminal region in protein complex formation

We considered the possibility that Ume6 may bind to DNA only indirectly through association with Efg1, Ndt80 or Upc2. To test this idea, we first analysed HA-tagged Ume6 deletion derivatives Ume6[1–720] and Ume6[1–759], which lack the C-terminal DNA binding domain (Fig. 4a). These derivatives retain a large N-terminal PrLD (Extended Data Fig. 2), a feature that in other TFs is critical for protein complex formation[6,23]. Strains expressing either deletion allele were defective in biofilm formation (Fig. 4b,c and Extended Data Fig. 2) and failed to form any of the three complexes (Fig. 4d). These data suggest that the Ume6 C-terminal region, which includes the DNA binding domain, is required for interaction with Efg1, Ndt80 and Upc2.

To test whether DNA binding ability is required for protein complex formation, we constructed mutant allele $UME6^{R772A}$, which is predicted to cause a DNA binding defect based on homology among zinc-cluster TFs[24–26]. Homozygous $UME6^{R772A}$ and $RBT5$-$UME6^{R772A}$ mutants were as biofilm defective as a $ume6\Delta/\Delta$ mutant (Fig. 4e,f), and $RBT5$-$UME6^{R772A}$ failed to rescue the biofilm and wrinkled colony defects of an $efg1\Delta/\Delta$ mutant (Extended Data Fig. 3). Therefore, Ume6(R772A) is functionally defective. To assay DNA binding by Ume6(R772A), we conducted ChIP-qPCR with the $ECE1$ and $HWP1$ promoter regions (Fig. 4g). The R772A substitution abolished Ume6 binding to these promoters. We assessed Ume6(R772A) complex formation with a co-IP assay and observed that it retained the ability to interact with Efg1, Ndt80 and Upc2 (Fig. 4h). These results indicate that Ume6 DNA binding ability is not required for complex formation with partner TFs.

To determine whether the C-terminal region is sufficient for protein complex formation, we analysed strains expressing HA-tagged Ume6[720–843], which harbours the DNA binding domain and short flanking segments (Fig. 4a). Strains homozygous for this allele, expressed from its native promoter or the $RBT5$ promoter, were as biofilm defective as a $ume6\Delta/\Delta$ mutant (Fig. 5a,b). ChIP-qPCR analysis showed that Ume6[720–843] retains DNA binding ability at the $ECE1$ promoter (Fig. 5c). Co-IP assays indicated that Ume6[720–843] retains the ability to form protein complexes with Efg1, Ndt80 and Upc2 (Fig. 5d). Taken together, these findings show that the C-terminal region of Ume6 is necessary and sufficient for protein complex formation.

## Impact of partner TFs on Ume6 gene regulation

To define the impact of Ume6-interacting TFs on Ume6 function, we tested strains overexpressing $UME6$ in individual $efg1\Delta/\Delta$, $ndt80\Delta/\Delta$ and $upc2\Delta/\Delta$ mutant backgrounds for biofilm formation, Ume6 DNA binding and Ume6 target gene activation. All three TF mutants presented biofilm defects under moderate inducing conditions for hyphae and biofilm (yeast extract–peptone–dextrose (YPD) + 400 μM bathophenanthrolinedisulfonic acid (BPS) media) (Fig. 6a). Although Upc2 has not been reported previously to be required for biofilm formation, we observed that the $upc2\Delta/\Delta$ mutant is defective in producing hyphae (Extended Data Fig. 4a,b,e) and biofilm (Extended Data Fig. 4c,d) in several growth conditions. In addition, the $upc2\Delta/\Delta$ mutant is biofilm defective in vivo in a rat venous catheter biofilm model (Extended Data Fig. 4f). Moreover, overexpression of $UME6$ in each mutant restored biofilm formation ability (Fig. 6a). These results indicate that all three Ume6-interacting TFs are required for biofilm formation and that Ume6 retains the ability to activate biofilm-relevant genes in the absence of any one interacting partner.

We tested the impact of each TF on Ume6–DNA binding using ChIP-qPCR in YPD + 400 μM BPS medium, again using 4-h planktonic hyphal growth to represent early stages of biofilm formation. We first focused on promoters of hypha-associated genes $ECE1$, $HWP1$, $HGC1$ and $HYR1$, all of which are bound by Ume6 in a wild-type background (Fig. 6b). Absence of Efg1 caused significantly reduced Ume6 binding (Fig. 6b) at all four promoter regions. Absence of Ndt80 caused significantly reduced Ume6 binding at the $ECE1$ and $HGC1$ promoters (Fig. 6b) and caused reduced expression (Fig. 6d,e). Absence of Upc2 did not affect Ume6 binding to these promoters (Fig. 6b). We turned to Upc2 target genes[27,28] and found that they include the biofilm-associated metabolic gene $ERG251$ (refs. 27,28), which is also bound by Ume6 (Extended Data Fig. 4g) and is required for biofilm formation[27]. Absence of Upc2 caused significantly reduced Ume6 binding at the $ERG251$ promoter (Fig. 6c) and reduced activation by overexpressed $UME6$ (Fig. 6f). We also tested the convergence of Ume6 and Upc2 impact under biofilm growth conditions, with assays of RNA levels for ergosterol genes, hypha-associated genes and hypoxia-responsive genes[29,30]. Among 14 genes tested that were dependent upon Upc2, we observed that Ume6 is also required for full expression of 12 genes (Fig. 6g,h). Overall, these data show that Efg1, Ndt80 and Upc2 are required for full levels of Ume6 binding to and activation of biofilm-relevant genes.

## Discussion

Ume6 is functionally distinct from many *C. albicans* biofilm regulators because its expression is induced alongside hypha-associated genes, and because engineered $UME6$ overexpression can bypass numerous regulatory and environmental requirements for biofilm and hypha formation[7,9,10]. Our findings here explain why Ume6 is so effective in driving biofilm and hypha formation: its direct targets include numerous genes known to be required for these processes. Surprisingly, though, Ume6 DNA binding is highly dependent upon three regulators—Efg1, Ndt80 and Upc2—a dependence that is explained by their interaction with Ume6. Ume6 may be considered

---

**Fig. 4 | The Ume6 C-terminal region is essential for protein–protein interaction. a**, Diagram depicting Ume6 mutants. The green boxes indicate zinc-cluster DNA binding domains (DBDs), the blue boxes indicate 3× HA tag and the red line indicates an amino acid substitution (arginine to alanine). **b**, For biofilm formation, cells were grown in RPMI + 10% FBS at 37 °C for 24 h, and biofilms were stained with calcofluor-white. The side-view projection images represent three independent experiments (*n* = 3). **c**, Bar graph indicating biofilm volume measurements for each strain. Biofilm volume measurements were performed with three biological independent samples using ImageJ (*n* = 3). The values are the mean ± s.d. of three biological replicates. Statistical significance was determined using one-way ANOVA (Dunnett's multiple-comparison test). Three biological replicates were used. **d**, Cells were grown in RPMI + 10% FBS at 37 °C for 4 h, and protein lysates from strains expressing the indicated HA- and/ or FLAG-tagged protein were subjected to immunoprecipitation with anti-HA antibody. The precipitated proteins were detected using either anti-HA or anti-FLAG for western blot analysis. The cell lysates used for co-IP were used as an input control. The asterisks indicate non-specific bands. The images represent three independent experiments (*n* = 3). **e**, For biofilm formation, cells were grown in RPMI + 10% FBS at 37 °C for 24 h, and biofilms were stained with calcofluor-white. The side-view projection images represent three independent experiments (*n* = 3). **f**, Bar graph indicating biofilm volume measurements for each strain. Biofilm volume measurements were performed with three biological independent samples using ImageJ (*n* = 3). Values are the mean ± s.d. of three biological replicates. Statistical significance was determined using one-way ANOVA (Dunnett's multiple-comparison test). **g**, ChIP-qPCR was performed using IP and input DNA from the WT (untagged), Ume6-HA and Ume6(R772A)-HA strains grown in RPMI + 10% FBS at 37 °C for 4 h. Values are the mean ± s.d. of three biological replicates. Statistical significance was determined using two-sided unpaired Student's *t*-test. The images represent three independent experiments (*n* = 3). **h**, Cells were grown in RPMI + 10% FBS at 37 °C for 4 h, and protein lysates from strains expressing the indicated HA- and/or FLAG-tagged protein were subjected to immunoprecipitation with anti-HA antibody. The precipitated proteins were detected using either anti-HA or anti-FLAG for western blot analysis. The cell lysates used for co-IP were used as input control. The asterisk indicates a non-specific band. The images represent three independent experiments (*n* = 3). *P* < 0.0001; the exact *P* values were <0.0001 for all indicated comparisons. All strains used in this figure are homozygous for alleles indicated. The genotype of the untagged strain is $UME6^{+/+}$.

a hitchhiker that interacts with other TFs to reach its destination (Fig. 6i). An interaction-driven DNA binding mechanism is well suited for a regulator of hyphal extension such as Ume6, because it overlays Ume6-dependent activation on previously selected target genes. Moreover, the interleaved binding profiles of Efg1 and Ndt80 (ref. 21) suggest an explanation for the ability of Ume6 to promote biofilm and hypha formation in the absence of Efg1 or its upstream regulatory signals (discussed below). Finally, we found that Ume6 extends the functional repertoire of the biofilm regulatory network,

for example, by binding to ergosterol biosynthesis gene promoter regions. This binding occurs in part through association of Ume6 with *ERG* gene activator Upc2. We propose that Ume6 maintains biofilm network target expression through association with Efg1 and Ndt80, and expands the network through association with Upc2.

Recent studies show that Efg1 and most other biofilm master regulators have PrLDs that mediate assembly into protein complexes[6,23]. These complexes may form phase-separated condensates, more transient hubs or both[6,23]. The PrLDs, where tested, are required for

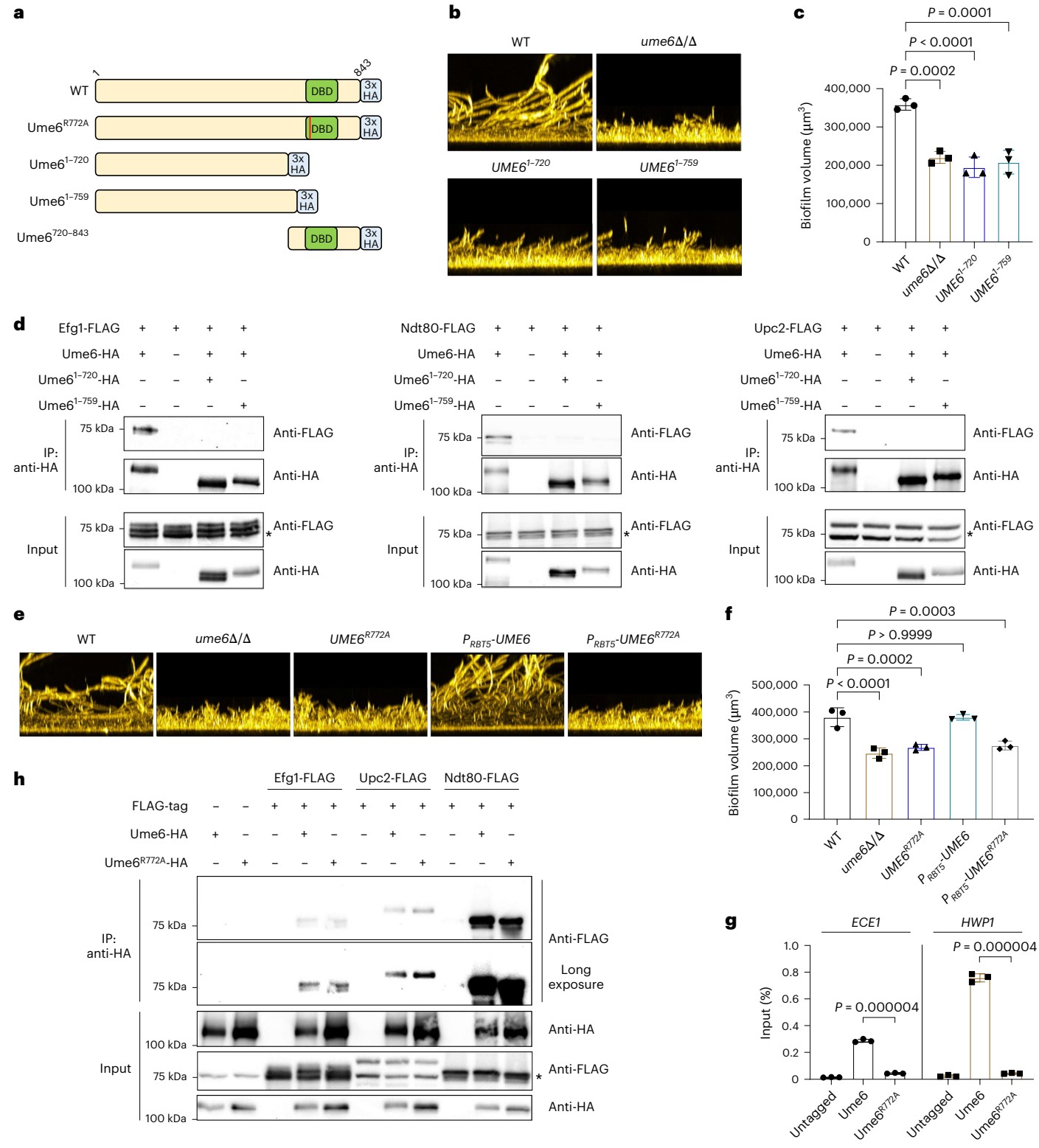

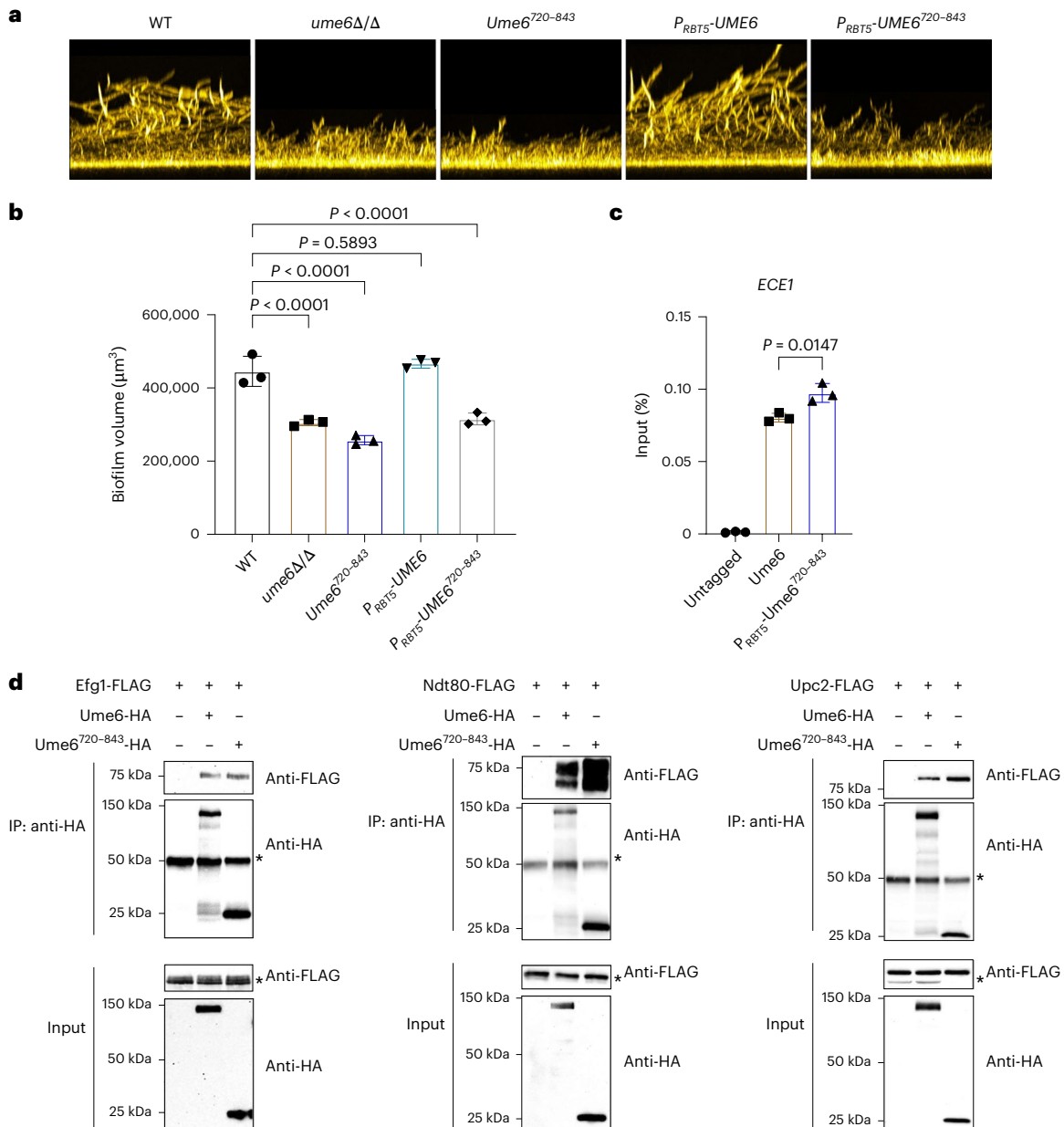

**Fig. 5 | The Ume6 C-terminal region is sufficient for protein–protein interaction. a**, For biofilm formation, cells were grown in RPMI + 10% FBS at 37 °C for 24 h, and biofilms were stained with calcofluor-white. The side-view projection images represent three independent experiments ($n = 3$). **b**, Bar graph indicating biofilm volume measurements for each strain. Biofilm volume measurements were performed with three biological independent samples using ImageJ ($n = 3$). Values are the mean ± s.d. of three biological replicates. Statistical significance was determined using one-way ANOVA (Dunnett's multiple-comparison test). Three biological replicates were used. **c**, ChIP-qPCR was performed using three biological replicates grown in RPMI + 10% FBS medium for 4 h at 37 °C ($n = 3$). ChIP was performed using formaldehyde-fixed cell lysates against anti-HA antibody. Values are the mean ± s.d. of three biological replicates. Statistical significance was determined using two-sided unpaired Student's *t*-test. **d**, Cells were grown in RPMI + 10% FBS at 37 °C for 4 h, and protein lysates from strains expressing the indicated HA- and/or FLAG-tagged protein were subjected to immunoprecipitation with anti-HA antibody. The precipitated proteins were detected using either anti-HA or anti-FLAG for western blot analysis. The cell lysates used for co-IP were used as an input control. The asterisks indicate non-specific bands. The images represent three independent experiments ($n = 3$). $P < 0.0001$; the exact $P$ values were <0.0001 for all indicated comparisons. All strains used in this figure are homozygous for alleles indicated. The genotype of the untagged strain is $UME6^{+/+}$.

function, thus making a compelling case for biological importance of the complexes[6,23]. Ume6 protein–protein interactions are distinct from PrLD-mediated TF assemblies in two ways. First, the large Ume6 N-terminal region, which includes multiple PrLDs, is dispensable for complex formation with Efg1, Ndt80 and Upc2. Second, whereas DNA binding domains may be dispensable for PrLD-mediated TF assemblies, we found that the Ume6 DNA binding domain and short flanking regions, carried in Ume6$^{720-843}$, are sufficient for complex formation.

For these reasons, we infer that Ume6 association with partner TFs is not through PrLD–PrLD interaction.

The DNA binding ability of Ume6 is not required for complex formation, as shown by the properties of Ume6(R772A). Ume6(R772A) properties also indicate that Ume6 cannot be stably targeted to DNA solely by its complex with partner TFs. These observations, together with the lack of a clear Ume6-associated DNA motif, suggest that Ume6 may have a low affinity for its DNA target sites. Hence, a more

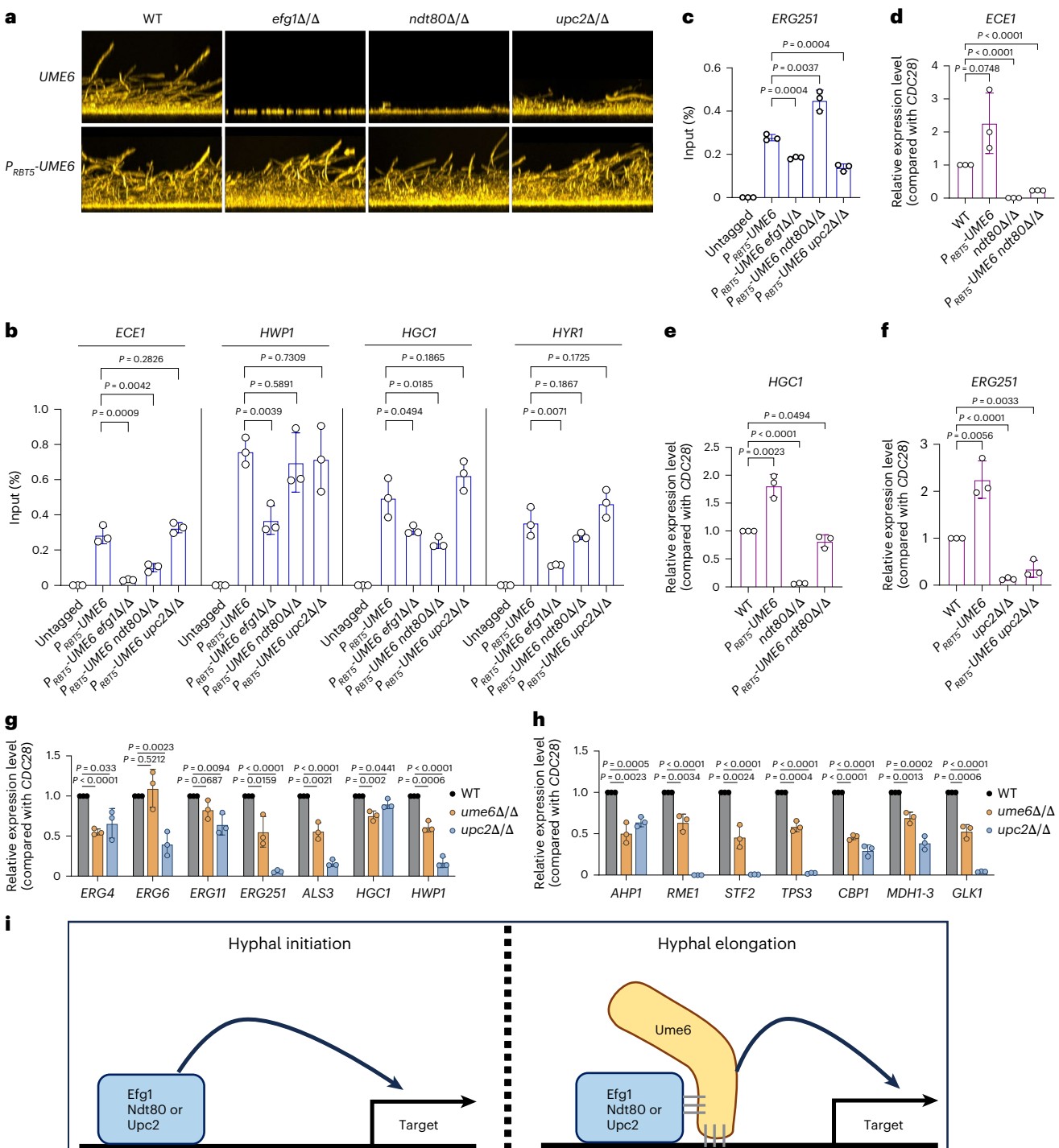

**Fig. 6 | Partner TFs Efg1, Ndt80 and Upc2 affect Ume6 function. a**, For biofilm formation, cells were grown in YPD + 400 µM BPS at 37 °C for 24 h, and biofilms were stained with calcofluor-white. The side-view projection images represent three independent experiments ($n = 3$). **b,c**, Ume6 partners Efg1, Ndt80 and Upc2 affect the DNA binding capability of Ume6. Data are presented for hypha-associated genes (**b**) or biofilm-associated metabolic gene ERG251 (**c**). ChIP-qPCR was performed using three biological replicates grown in YPD + 400 µM BPS for 4 h at 37 °C ($n = 3$). ChIP was performed using formaldehyde-fixed cell lysates against anti-HA antibody. Values are the mean ± s.d. of three biological replicates. Statistical significance was determined using two-sided unpaired Student's *t*-test. **d–f**, Bar graphs indicating relative *ECE1* (**d**), *HGC1* (**e**) or *ERG251* (**f**) mRNA levels in indicated strains. Each strain was grown in YPD + 400 µM BPS at 37 °C for 4 h. Total RNAs of three independent biological samples were extracted and used for qPCR determination ($n = 3$). Values are the mean ± s.d. of three biological replicates. Statistical significance was determined using two-sided unpaired

Student's *t*-test. **g,h**, Bar graphs indicating relative RNA levels from *ERG* genes, *ALS3*, *HGC1*, *HWP1* (**g**) and hypoxia-responsive genes (**h**) in indicated strains. Each strain was grown in YPD at 37 °C for 24 h under biofilm conditions. Total RNA from three independent biological samples was extracted and used for qPCR determination ($n = 3$). Values are the mean ± s.d. of three biological replicates. Statistical significance was determined using two-sided unpaired Student's *t*-test. All strains used in **a–h** are homozygous for alleles indicated. The genotype of the untagged strain is *UME6^{+/+}*. **i**, Hitchhiker model for Ume6 function. When hypha formation is initiated (left), *UME6* expression has just been induced and there is little Ume6 protein in the cell. Expression of newly induced genes is activated by Efg1, Ndt80 and Upc2 bound to their promoter regions. When hyphae reach the elongation stage (right), *UME6* has been fully induced. Ume6 reaches target gene promoters by hitchhiking with Efg1, Ndt80 or Upc2. We propose that Ume6 binds to DNA with low affinity, and binding is stabilized by Ume6–Efg1, Ume6–Ndt80 or Ume6–Upc2 complex formation.

stable tri-molecular complex of Ume6–DNA–partner TF may form only at high-affinity DNA binding sites for the partner TF, a mechanism that ensures that Ume6 activates genes pre-selected by the partner TFs.

We are not aware of other *C. albicans* TFs that function through low-affinity binding to DNA. However, in other eukaryotes, low-affinity TF binding has dramatic functional impact[31,32]. A TF hub, as formed by Efg1 and other biofilm master regulators, can increase the local concentration of TFs in the complex sufficiently to enable occupancy of low-affinity sites[31,32]. In this way, our proposal that Ume6 functions through low-affinity DNA binding sites fits well with the discovery of PrLD–PrLD interactions among biofilm master regulators and with the understanding of TF function in higher eukaryotes.

Is there a single complex of the three partner proteins with which Ume6 associates, or are there three separate Ume6–partner TF complexes? Efg1 has been shown to form PrLD–PrLD complexes with the TFs Wor1, Bcr1 and Flo8 (refs. 6,23); to our knowledge, Ndt80 and Upc2 have not been tested. Both Ndt80 and Upc2 have PrLD regions predicted by IUPred3 (ref. 33), so they may form PrLD–PrLD complexes. However, our data are consistent with recruitment of Ume6 by individual partner TFs. At *HWP1* and *HYR1*, binding of Ume6 depends upon Efg1 but not Ndt80 or Upc2; Efg1 alone may recruit Ume6. At *ECE1* and *HGC1*, binding of Ume6 depends upon both Efg1 and Ndt80, although not Upc2; Efg1 or Ndt80 may recruit Ume6. At *ERG251*, binding of Ume6 depends upon Upc2 much more than Efg1 or Ndt80; Upc2 may have the major role in recruiting Ume6. Thus, the simplest model that explains our data is that Ume6 can be recruited by Efg1, Ndt80 or Upc2. Although Ume6 may also bind to some regions independently of any partner TFs, we have not found an example as of yet.

Ume6 orthologs activate filamentation in many pathogenic *Candida* species: *UME6* ortholog expression is induced early during filamentation, and its overexpression drives increased filamentation. Two observations suggest that these Ume6 orthologs use a partner-driven gene activation mechanism that is similar to that of *C. albicans* Ume6. First, *EFG1* orthologs are required for filamentation and biofilm formation in *Candida parapsilosis* and *Candida tropicalis*[34,35], and *UME6* ortholog overexpression in these species causes increased filamentation[12]. Hence, Efg1 and Ume6 are implicated in the same process. Second, in the case of *Candida auris*, *UME6* ortholog overexpression causes increased expression of ergosterol biosynthetic genes as well as efflux pump gene *MRR1* (ref. 11), all of which are targets of the *C. auris* Upc2 ortholog[36]. For these reasons, we suggest that Ume6–partner TF complexes permit Ume6 to hitchhike to target promoters in multiple *Candida* species.

The diversity of Ume6 binding partners provides an explanation for the ability of *UME6* expression to bypass many conventional hyphal and biofilm regulatory signals. Consider the biofilm adhesin genes *HWP1* and *HYR1*. Both are bound and activated by Efg1. When Ume6 is expressed, it also binds to their promoter regions. Ume6 binding is partially Efg1 dependent, a suggestion that it is bound in a Ume6–Efg1 complex. If Ume6 is expressed in the absence of Efg1, it still has some ability to bind to those promoter regions. Both promoter regions have Upc2 and Ndt80 motifs. Therefore, in the absence of Efg1, Ume6 may be recruited to those regions by forming Ume6–Ndt80 or Ume6–Upc2 complexes. In the case of the hyphal cyclin gene *HGC1*, both Efg1 and Ndt80 motifs are present, so Ume6 may be recruited in Ume6–Efg1 or Ume6–Ndt80 complexes. Thus, the potency of Ume6 arises from its ability to form complexes with multiple partners that bind to many of the same regulatory regions. In essence, Ume6 can hitchhike to the same destination by riding with different partners.

Ume6 has seemed distinct from biofilm master regulators because it is required to maintain, not initiate, expression of genes for biofilm and hypha formation[7]. Our results here show that Ume6 function aligns with that of the biofilm master regulators in several ways. First, its direct targets include major biofilm determinants and hypha-associated

genes, such as *ALS1*, *ALS3*, *ECE1*, *HGC1*, *HWP1*, *HYR1*, *SOD5* and *SOD6*. Second, its direct targets also include the biofilm master regulator genes *BCR1*, *BRG1*, *EFG1*, *FLO8*, *RFX2* and *TEC1*. Third, its direct targets include additional TF genes that influence filamentation or biofilm formation under particular environmental conditions, including *ZAP1/CSR1*, *RIM101*, *TYE7* and *WOR3*. In fact, Ume6 binds to all eight genes that are shared targets of the biofilm master regulators[21]. These observations indicate that Ume6 is tightly integrated into the characterized biofilm regulatory network. This integration includes both Ume6 function and interaction: it forms complexes with two biofilm master regulators, Efg1 and Ndt80. Ume6 complex formation sets it apart from the biofilm master regulators in that its interaction region is close to or within its DNA binding domain and independent of its large PrLD region. Moreover, its interacting partners include hypoxic response regulator Upc2, an interaction that contributes to the expression of biofilm metabolic gene *ERG251* and probably promotes expression of several hypoxic response genes during biofilm formation. In this way, Ume6 may be viewed as a bridge that connects the hyphal morphogenesis and adherence genes that determine biofilm architecture, and the hypoxic response genes necessary for growth in the low-oxygen biofilm environment.

## Methods

### Strains and media

Strains used in this study were maintained in 15% glycerol frozen stocks at −80 °C. Before use, cells were routinely grown on YPD agar plates (2% dextrose, 2% Bacto peptone, 1% yeast extract, 2% Bacto agar) overnight at 30 °C and then cultured in liquid YPD medium overnight at 30 °C with agitation. Transformants were selected on YPD plus 400 µg ml⁻¹ nourseothricin (clonNAT; Gold Biotechnology) or complete synthetic medium (2% dextrose, 1.7% Difco yeast nitrogen base with ammonium sulfate and auxotrophic supplements). All strains used in this study are listed in Supplementary Table 4.

### Strain constructions

To manipulate the *C. albicans* genome, the transient clustered regularly interspaced short palindromic repeat (CRISPR) and CRISPR-associated gene 9 (CRISPR-Cas9) system was used as previously described in detail[37]. Generally, the Cas9 cassette was amplified from the plasmid pV1093, and each single guide RNA (sgRNA) cassette was generated by using split-joint PCR with 'sgRNA/F YFG1' and 'SNR52/R YFG1' as previously described in detail[16,37]. Primers and plasmids used for transformation are listed in Supplementary Table 5.

To construct the Ume6-HA-tagged strain in *C. albicans* isolates, the plasmid pED3-HA was used for amplification of the 3× HA tag cassette containing the *CdHIS1* marker with primers 'Ume6_F-HA' and 'Ume6_R-HA'[18]. In the *his1*Δ/Δ mutant of each isolate background, Ume6 downstream regions, 303 bp (SC5314) or 315 bp (P75010), were replaced with the Ume6-HA cassette by using Cas9 and Ume6term sgRNA cassette. Transformation was conducted with approximately 2 µg of Cas9, 1 µg of sgRNA and 3 µg of the Efg1-HA repair template. Transformants were screened on complete supplement mixture (CSM) lacking histidine plate, and candidates were genotyped by PCR using primers 'Ume6_int/F' and 'ACT1term int/R' for presence of the Ume6-tag cassette and using primers 'Ume6_int/F' and 'Ume6_tag confirm/R' for absence of the targeted *UME6* downstream region at the *UME6* locus. To test whether the *UME6-HA* alleles are functional, strains were assessed by hypha formation assays (Extended Data Fig. 1).

To construct the $P_{RBT5}$-*UME6* strain in the WT, *efg1*Δ/Δ, *ndt80*Δ/Δ and *upc2*Δ/Δ background, the repair template and sgRNA cassette were prepared as previously described in detail[17].

To construct strains expressing FLAG-tagged protein such as Efg1, Ndt80, Upc2 and Sef1 in an SC5314 background, a cassette containing the 3× FLAG tag, *ACT1* terminator, and either *CdHIS1* or *NAT* was amplified with pED1 or pED6, respectively, using primers containing

80 bp homology at the 3′ region for each gene. For Efg1, Ndt80, Upc2 and Sef1-FLAG-tagged strains, primers 'TF-FLAG/F' and 'TF-FLAG/R' were used for amplification of the FLAG tag cassette for each gene. Each tagged strain was verified using phenotypic assay (Extended Data Fig. 1).

To construct the strain expressing either Ume6$^{1-720}$-HA or Ume6$^{1-759}$-HA protein, a repair DNA cassette containing the 3× HA tag, *ACT1* terminator and marker gene was integrated at the 3′ region of Ume6. For the Ume6$^{1-720}$-HA cassette, primers 'Ume6_720 tag/F' and 'Ume6_R-HA' were used for amplification with pED4 (*CdHIS1*) or pED5 (*NAT*). For Ume6$^{1-759}$-HA, primers 'Ume6_759 tag/F' and 'Ume6_R-HA' were used for amplification with pED4 or pED5. Transformants were screened on CSM lacking histidine plate, and candidates were genotyped by PCR using primers 'Ume6_confirm int/F4' and 'ACT1term int/R' for presence of the Ume6-tag cassette and using primers 'Ume6_confirm int/F4' and 'Ume6_tag confirm/R' for absence of the targeted *UME6* downstream region at the *UME6* locus.

To construct the strain expressing Ume6$^{720-843}$-HA-tagged protein, a DNA cassette containing Ume6$^{720-843}$, 3× HA tag, *ACT1* terminator and *CdHIS1* gene was amplified with primers 'Ume6_720-843/F' + 'Ume6_3′UTR/R' using gDNA from the Ume6-HA-tagged strain. The amplified DNA cassette was replaced with the *NAT* marker gene at the original *UME6* locus in the *ume6*Δ/Δ mutant. Transformants were screened on CSM lacking histidine plate, and candidates were genotyped by PCR using primers 'Ume6_tag confirm/F' and 'ACT1term int/R' for the presence of the Ume6-tag cassette and using primers 'Ume6_tag confirm/F' and 'Ume6_tag confirm/R' for absence of the targeted *UME6* downstream region at the *UME6* locus.

Site-directed mutation at the zinc-cluster domain of Ume6 was conducted using PCR. Two fragments containing the R772A mutation were amplified with both pairs of primers 'Ume6 confirm/F' and 'Ume6_5′_R772A_R', and 'Ume6_5′_R772A_F' and 'Ume6_3′_UTR_R' using plasmid, pED24, which contains the Ume6 ORF, 3× HA, *ACT1* terminator and *CdHIS1* gene. The amplified fragments were integrated into the genome in *ume6*Δ/Δ mutant (*his1*Δ/Δ), and transformants were screened on CSM lacking histidine plate, and candidates were genotyped by PCR using primers 'Ume6_int/F' and 'ACT1term int/R' for presence of the Ume6-tag cassette and using primers 'Ume6_int/F' and 'Ume6_tag confirm/R' for absence of the targeted *UME6* downstream region at the *UME6* locus. The site-directed mutation was confirmed using Sanger sequencing.

To construct *ndt80*Δ/Δ mutant strains in any background, the *NDT80* deletion cassette was amplified from the plasmid pSFS2A-CaKan with primers 'Ndt80_deletion_Kan_F' and 'Ndt80_deletion_Kan_R'[38]. Transformants were screened on YPD containing 600 µg ml$^{-1}$ G418 and 1.75 mg ml$^{-1}$ molybdate, and candidates were genotyped by PCR using primers 'Ndt80 check up/F' and 'Ndt80 check int/R' for absence of *NDT80* ORF and using primers 'Ndt80 check up/F' and 'Kan int/R' for the presence of the *CaKAN* marker at the *NDT80* locus.

To construct *upc2*Δ/Δ mutant strains in any background, the *UPC2* deletion cassette was amplified from the plasmid pSFS2A-CaHygB with primers 'Upc2_deletion_HygB_F' and 'Upc2_deletion_HygB_R'[38]. Transformants were screened on YPD containing 600 µg ml$^{-1}$ hygromycin B and 1.75 mg ml$^{-1}$ quinine, and candidates were genotyped by PCR using primers 'Upc2 check up/F' and 'Upc2 check int/R' for absence of the *UPC2* ORF and using primers 'Upc2 check up/F' and 'HygB int/R' for the presence of the *CaHYGB* marker at the *UPC2* locus.

To construct the *UPC2* complementation strain in the *upc2*Δ/Δ mutant, the *UPC2* ORF region with the 3×FLAG-NAT cassette was integrated at the original locus. The *UPC2* ORF region was amplified with 'Upc2 check/F' and 'Upc2 ORF/R', and the UPC2-FLAG-NAT cassette was amplified with 'UPC2-FLAG/F' and 'UPC2-FLAG/R'. These two fragments were used for transformation with 1 µg of CaCas9 and 1 µg of KAN sgRNA for targeting the *KAN* marker at the *UPC2* original locus.

## Filamentation assay

To assay hyphal formation in *C. albicans* strains, cell culture and fixation were performed according to previous published methods[16]. Cells were grown in YPD overnight at 30 °C and transferred to an indicated pre-warmed medium with an OD of 0.5. Cells were incubated in a glass test tube for 4 h at 37 °C and fixed with 4% formaldehyde in phosphate-buffered saline (PBS) for 15 min. Calcofluor-white was used for cell staining; then cells were observed using a Zeiss Axio Observer Z.1 fluorescence microscope with a 20 × 0.8 numerical aperture.

## Biofilm assay

Biofilm production and imaging followed previously published methods with minor modifications[18,39]. Briefly, cells were grown in YPD overnight at 30 °C and the collected cells were washed with double-distilled water (ddH$_2$O). Then, cells were transferred to an indicated pre-warmed medium to achieve an OD$_{600}$ of 0.05 in a 96-well plate (Greiner, 655090) and incubated at 37 °C for 90 min for cell adherence. Non-adherent cells were removed by washing with PBS; the indicated pre-warmed medium was added to each well and incubated for 24 h at 37 °C with shaking at 60 rpm. Biofilms were fixed with 4% formaldehyde, washed with PBS and stained with calcofluor-white (200 µg ml$^{-1}$ in PBS); then 2,2′-thiodiethanol in PBS was added to each well for clarification and refractive index matching. Each biofilm sample was prepared as a biological triplicate and observed using a Keyence fluorescence microscope (BZ-X800E) with a Keyence ×20 objective and ×2 zoom.

## RNA extraction and sequencing

Cells were grown in YPD at 30 °C overnight and washed with ddH$_2$O once. For RNA extraction from planktonic conditions, cells were inoculated into 25 ml of RPMI 1640 medium + 10% foetal bovine serum (RPMI + 10% FBS) or YPD + 400 µM BPS medium to an OD$_{600}$ of 0.2 and incubated at 37 °C for 4 h with shaking at 225 rpm. For RNA extraction from biofilm conditions, cells were inoculated into YPD medium to an OD$_{600}$ of 0.2 and incubated at 37 °C for 24 h without agitation. Then cells were collected using vacuum filtration and quickly frozen at −80 °C. RNA extraction, library preparation and RNA sequencing were performed according to previously described methods[16].

## qRT-PCR

RNA samples were prepared from three biological replicates of each strain, and 1 µg of total RNA per sample was used for cDNA synthesis using an iScript gDNA Clear cDNA Synthesis Kit (Bio-Rad, 1725035) with diluted cDNA, following the manufacturer's instruction. Then, qRT-PCR was conducted using the iQ SYBR Green Supermix (Bio-Rad, 1708880). The *CDC28* gene was used for endo-control gene to normalize tested genes in each experiment using the threshold cycle ΔΔ$C_T$ method.

## ChIP assay

ChIP was performed according to previously described methods[18]. To extract soluble chromatin from *C. albicans*, strains grown in 5 ml YPD at 30 °C overnight were inoculated into 100 ml of RPMI + 10% FBS to an OD$_{600}$ of 0.2. Cells were then cultured at 37 °C for 4 h in a shaking incubator with 225 rpm, fixed with formaldehyde (1% final concentration) for 15 min at RT and quenched with glycine (300 mM final concentration) for 10 min at RT. Cells were washed with ice-cold PBS twice and lysed with FA lysis buffer (50 mM HEPES–KOH, 140 mM NaCl, 1 mM EDTA, 1% Triton X-100, 0.1% sodium deoxycholate, 0.1% SDS, 1 mM PMSF and 1× proteinase inhibitor cocktail) and glass beads using a bead beater. Cell lysates were clarified by centrifugation and sonicated using a Bioruptor sonicator (Diagenode) for 30 cycles. Then the lysates were clarified by centrifugation and used for immunoprecipitation as a previously described method with minor modifications[40]. Briefly, Dynabeads protein G (Invitrogen, 10003D) was conjugated with anti-HA antibody (Abcam, ab9110) and incubated with sheared chromatin samples at 4 C overnight with rotation. The beads were washed three

times with FA lysis buffer, three times with FA lysis buffer containing 500 mM of NaCl and one time with LiCl buffer (0.25 M LiCl, 1% sodium deoxycholate, 1 mM EDTA, 10 mM Tris, pH 8.0) using magnet strand (Invitrogen). Finally, the beads were washed with TE (10 mM Tris–HCl, pH 8.0, and 1 mM EDTA) and resuspended with elution buffer (50 mM Tris–HCl, pH 8.0, 10 mM EDTA and 1% SDS). The beads were then incubated at 65 °C for 15 min, and eluted samples were incubated at 65 °C overnight. Each sample was treated with proteinase K (Invitrogen) and RNaseA (Invitrogen) at 50 °C for 1.5 h, and reverse-cross-linked DNA samples were purified using the ethanol precipitation method with glycogen (Invitrogen). DNA quantification was performed by using a Qubit dsDNA HS Assay Kit (Molecular Probes).

## ChIP-qPCR
ChIP-qPCR was conducted with immunoprecipitated DNA and input DNA from Ume6-HA-tagged strains using the iQ SYBR Green Supermix (Bio-Rad, 1708880), following the manufacturer's instruction.

## ChIP-seq
ChIP-seq libraries were prepared according to previously described methods[18]. Libraries were prepared in biological triplicate from each strain using the NEBNext Ultra II DNA Library Prep Kit for Illumina (New England Biolabs) according to the manufacturer's instructions with the following modifications. For input DNA samples, 75 ng of DNA was used as starting template for the libraries. The adaptors were diluted 1:10 and 7 PCR amplification cycles were performed. For IP DNA samples, the libraries were prepared with 1.6 ng of DNA as starting template, the adaptors were diluted 1:25 and 12 PCR cycles were performed to amplify the library. The IP libraries were size selected by adding 36 µl of nuclease-free water to 14 µl of library. Agencourt AMPure XP beads (Beckman Coulter) were added to a final volume of 75 µl, and after magnetization, the supernatant was transferred to a new tube containing 42.5 µl of beads. After magnetization, the pellet was twice washed with 80% EtOH and the library was eluted in 15 µl of nuclease-free water. The libraries were pooled and sequenced (2 × 51 nt, paired end) on a NextSeq 2000 Sequencing System (Illumina).

## Western blot assay
Western blot assay was performed according to previously described methods[18]. Briefly, cells were grown in YPD at 30 °C overnight and washed with ddH$_2$O once. Cells were then inoculated into RPMI + 10% FBS to an OD$_{600}$ of 0.2 and incubated for 4 h at 37 °C with shaking at 225 rpm (planktonic conditions) or incubated for 24 h at 37 °C without shaking (biofilm conditions). Total protein was prepared using bead beating with FA lysis buffer (50 mM HEPES-KOH, 140 mM NaCl, 1 mM EDTA, 1% Triton X-100, 0.1% sodium deoxycholate, 1 mM PMSF and 1× proteinase inhibitor cocktail) with glass beads (Sigma). The cell debris was removed using centrifugation at 16,200 × g for 5 min at 4 °C, and protein concentrations were measured using Bradford assay (Bio-Rad). A total of 40 µg of soluble protein from each strain was loaded and separated into 8% or 10% SDS polyacrylamide gel and transferred to nitrocellulose membrane (Bio-Rad). Then, transferred proteins on the membrane were visualized using Ponceau S (Sigma) for confirming equal protein loading in each well. For western blot analysis, each membrane was blocked with 5% non-fat dry milk (Bio-Rad) in TBST (tris-buffered saline containing 0.05% Tween 20, pH 7.4) overnight at 4 °C and washed three times with TBST. Anti-HA polyclonal rabbit antibody (Abcam; ab9110) or anti-FLAG polyclonal mouse antibody (Sigma; F3165) was used as primary antibody, and goat anti-mouse IgG (H + L) secondary antibody, HRP (Thermo Fisher, catalogue number 31430) or anti-rabbit IgG-HRP from goat (Jackson ImmunoResearch; 111-035-144) were used as secondary antibody. Fluorescent signals were developed using enhanced chemiluminescence (Thermo Fisher Scientific, 34580) and imaged using ChemStudio (Analytik Jena).

## Co-immunoprecipitation assay
Cells were grown in YPD at 30 °C overnight and washed with ddH$_2$O once. For protein extraction from planktonic conditions, cells were inoculated into 25 ml of RPMI + 10% FBS to an OD$_{600}$ of 0.2 and incubated at 37 °C for 4 h with shaking at 225 rpm. For protein extraction from biofilm conditions, cells were inoculated into 25 ml of RPMI + 10% FBS to an OD$_{600}$ of 0.2 and incubated at 37 °C for 24 h without agitation. Cultured cells were washed with ice-cold PBS twice, and whole-cell proteins were extracted with glass beads and FA lysis buffer containing proteinase inhibitor cocktail, phosphatase inhibitor cocktail, and PMSF by using a mini bead beater. A total of 2 mg of whole-cell lysate from each strain was incubated with 1 µg of polyclonal anti-HA antibody (Abcam; ab9110) overnight at 4 °C with agitation, then the antigen–antibody complex was pulled down by adding Dynabeads protein G (Invitrogen, 10003D) for 3 h at 4 °C. The antigen–antibody-complex-bound Dynabeads were washed with FA lysis buffer twice, eluted with 1× SDS-sample buffer and subjected to SDS-PAGE for western blot analysis.

## Bioinformatic analysis
Bioinformatic analysis used in this study was performed according to previously described methods[18]. Briefly, raw Illumina FASTQ data from all strains and samples were aligned to the *C. albicans* genome release 21 using bowtie2 (v2.1.0; default options). The resulting sam-formatted alignment files were converted to bam format, sorted and indexed using samtools (v1.10). Aligned bam files were filtered to retain only proper-paired alignments (samtools view -f 0×02 -b -h) and subsampled to ~10 M reads (samtools -s -b -h). ChIP-seq peaks were called using MACS2 (v 2.1.0.20151222; default parameters). For SC5314 and P75010, the untagged IP samples were used as controls. narrowPeak files from each replicate (3 per strain) were combined using mspc (v 5.4.0; options -r bio -w 1e-4 -s 1e-8) to define reproducible consensus peaks. MACS2 peak summits from each replicate were concatenated, then bedTools was used to sort and merge (-d 20) overlapping summits. MACS2 peak summits were mapped to mspc consensus peaks using intersectBed and annotated to protein coding genes using closestBed (bedTools v 2.17.0). To eliminate false positives from the Ume6 ChIP-seq data, the peaks that are uniformly enriched in both Ume6 IP and untagged control were discovered by visual inspection and discarded from the peak annotation data. Differential binding analysis of ChIP-seq peaks between strains was conducted using DiffBind (Galaxy Version 2.10.0+galaxy0) with default parameters[41].

For de novo motif discovery, DNA sequences within 400 bp of peak centres were analysed by using HOMER (v 4.11) findMotifsGenome.pl with options '-size 400 -mask'[42]. In total, 429 binding sites were selected based on the binding score and *q*-value. Extract Genomic DNA (Galaxy Version 3.0.3+galaxy2) was used for extraction of desired genomic DNA sequences[43]. NCBI BLAST + blastn (Galaxy Version 2.10.1+galaxy0) with 0.001 cutoff was used for finding homologies between SC5314 and clinical isolates. BamCoverage (Galaxy Version 3.3.2.0.0) with 20 bp of bin size was used for visualization of transcriptome data in the Integrative Genomics Viewer. ComputeMatrix (Galaxy Version 3.5.4+galaxy0) with '--beforeRegionStartLength 2000 --afterRegionStartLength 2000 --binSize 10' and plotHeatmap (Galaxy Version 3.5.4+galaxy0) were used for generating a heatmap for ChIP-seq results. Prediction of activation domain was performed using PADDLE[44]. Interpretations and hypotheses were always guided by the comprehensive information at the *Candida* Genome Database[45] and FungiDB[46]. GO term enrichments were determined with the GO Termfinder tool at the *Candida* Genome Database.

## Data analysis software
ChIP-seq data were visualized using the Integrative Genomics Viewer v2.11.0 (ref. 47). Biofilm and filamentation images were processed using ImageJ (Fiji)[48]. Statistical analyses and graph generations were carried out using GraphPad Prism version 9 (GraphPad software). Volcano

plots were generated using VolcaNoseR (https://huygens.science.uva.nl/VolcaNoseR2/). IUPRED3 (https://iupred3.elte.hu/) software were used for predicting IDRs in each TF.

## Ethics statement

All animal procedures were approved by the Institutional Animal Care and Use Committee at the University of Wisconsin-Madison according to the guidelines of the Animal Welfare Act, the Institute of Laboratory Animal Resources Guide for the Care and Use of Laboratory Animals and Public Health Service Policy. The approved animal protocol number is DA0031.

## Reporting summary

Further information on research design is available in the Nature Portfolio Reporting Summary linked to this article.

## Data availability

Processed RNA-seq and ChIP-seq data are available in Supplementary Tables 1 and 2 respectively; raw data are available through NCBI SRA with accession numbers PRJNA1114072 (RNA-seq), PRJNA1114713 (Ume6-HA ChIP-seq) and PRJNA1114694 (Ume6-HA overexpression ChIP-seq). Strains are available upon request. Source data are provided with this paper.

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

## Acknowledgements

We thank M. Kuhr for lab management and technical support, F. Lanni and Mitchell lab members for their continued interest and many helpful discussions, and X. Lin, Z. Lewis and M. Momany, for advice and support. This work was supported by NIH grants R01 AI146103 (A.P.M.) and R01 AI073289 (D.R.A.) and by Distinguished Research Professorship funds from the University of Georgia (A.P.M.).

## Author contributions

E.D., C.J.M., D.R.A. and A.P.M. designed the experiments. E.D. and R.Z. performed the experiments. E.D., C.J.M. and D.R.A. analysed the data. M.Y.H. and K.G. constructed and validated *C. albicans* strains. E.D. and A.P.M. wrote the paper.

## Competing interests

The authors declare no competing interests.

## Additional information

**Extended data** is available for this paper at https://doi.org/10.1038/s41564-025-02094-5.

**Correspondence and requests for materials** should be addressed to Aaron P. Mitchell.

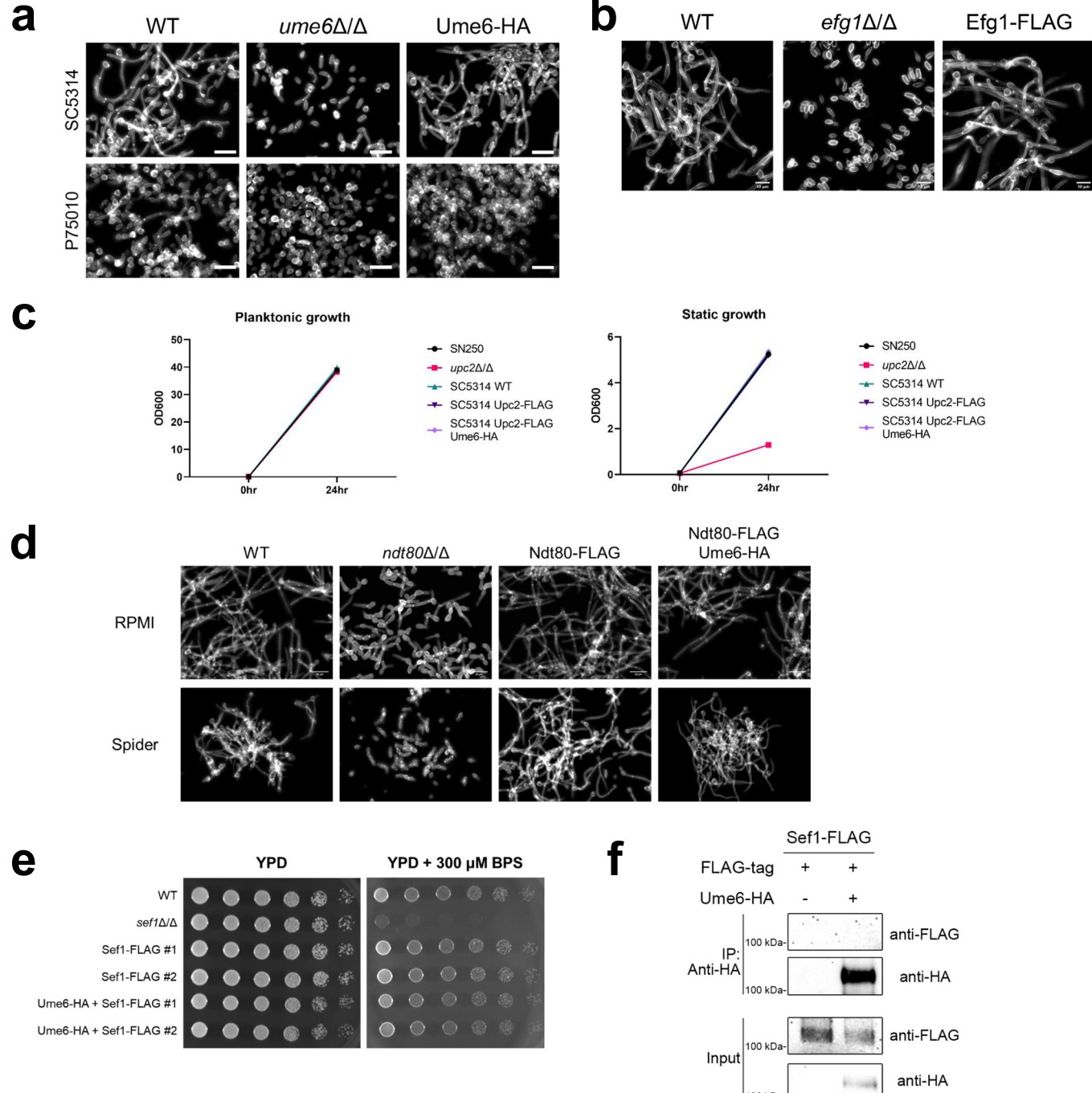

**Extended Data Fig. 1 | Validation of HA tagged strains. (a)** SC5314 strains were grown in YPD + 10% FBS medium at 37 °C for 4 h. P75010 strains were grown in RPMI + 10% FBS medium at 37 °C for 4 h. Fixed cells were stained with Calcofluor-White and imaged using fluorescence microscope. Scale bar indicates 20 microns. The images represent three independent experiments. **(b)** SC5314 strains were grown in RPMI + 10% FBS medium at 37 °C for 4 h. Fixed cells were stained with Calcofluor-White and imaged using fluorescence microscope. Scale bar indicates 10 microns. The images represent three independent experiments. **(c)** Cells were grown in YPD medium under planktonic or static growth condition at 30 °C. **(d)** Filamentation assay under either RPMI or Spider media at 37 °C for 4 h. Cells were fixed with formalin and stained with Calcofluor-White. White

bar indicates 20 microns. The images represent three independent experiments. **(e)** Strains were grown in YPD at 30 °C for overnight, and serial diluted cells (5-fold dilutions from OD$_{600}$ ~ 3) were spotted onto YPD agar medium with or without 300 μM BPS. The plates were incubated at 30 °C for 24 h and imaged. **(f)** Cells were grown in RPMI + 10% FBS at 37 °C for 4 h, and protein lysates from strains expressing the indicated HA- and/or FLAG-tagged protein were subjected to immunoprecipitation with anti-HA antibody. The precipitated proteins were detected using either anti-HA or anti-FLAG for Western blot analysis. The cell lysates used for co-IP were used as an input control. The images represent three independent experiments. All strains used in this figure are homozygous for alleles indicated. The WT genotype is *UME6-HA/UME6-HA*.

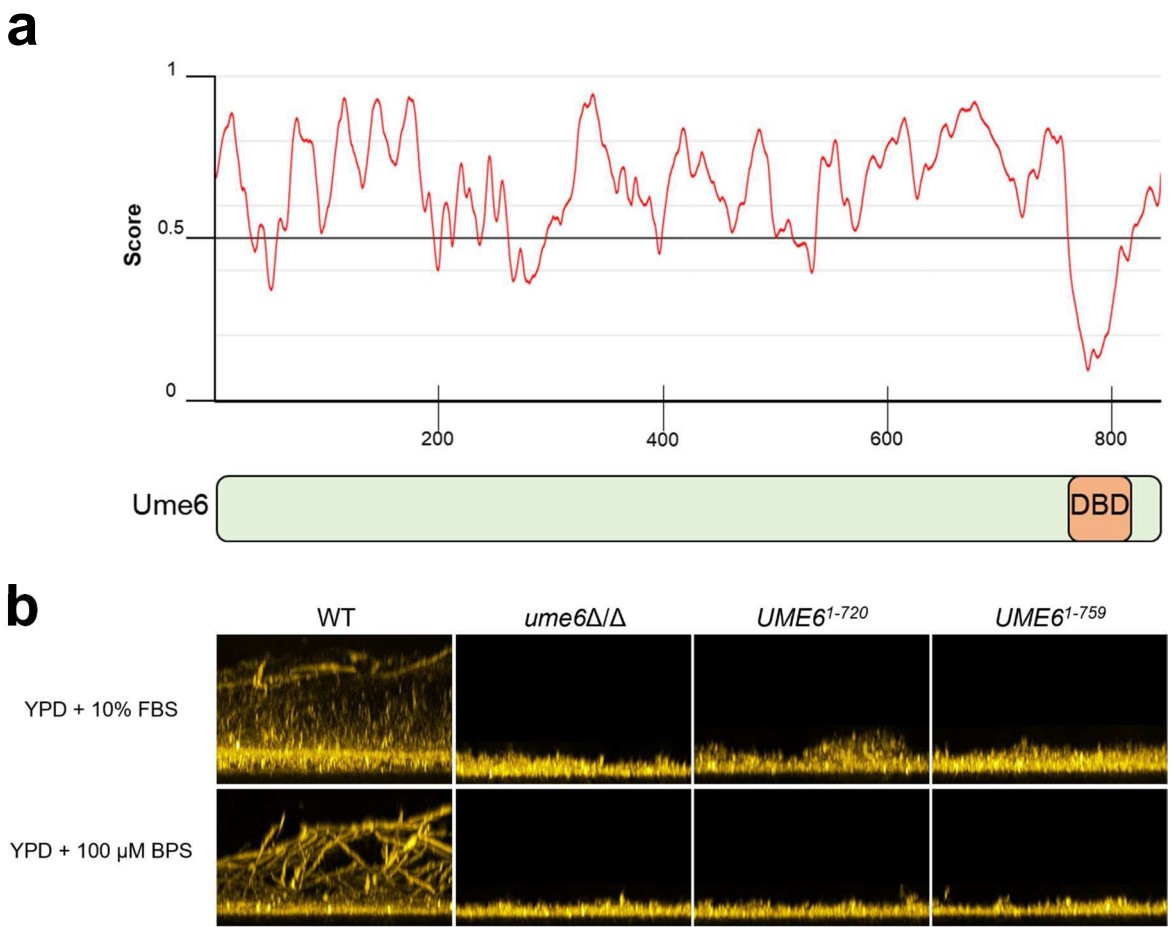

**Extended Data Fig. 2 | Ume6 protein contains PrLDs. (a)** The prediction of intrinsically disordered regions (prion-like domains; PrLDs) of Ume6 was conducted using the IUPRED3 (https://iupred3.elte.hu/) software. **(b)** Biofilm formation under YPD + 10% FBS or YPD + 100 μM BPS for 24 h at 37 °C. Biofilms were stained with Calcofluor-White. The side-view projection images represent three independent experiments. All strains used in this figure are homozygous for alleles indicated. The WT genotype is *UME6-HA/UME6-HA*.

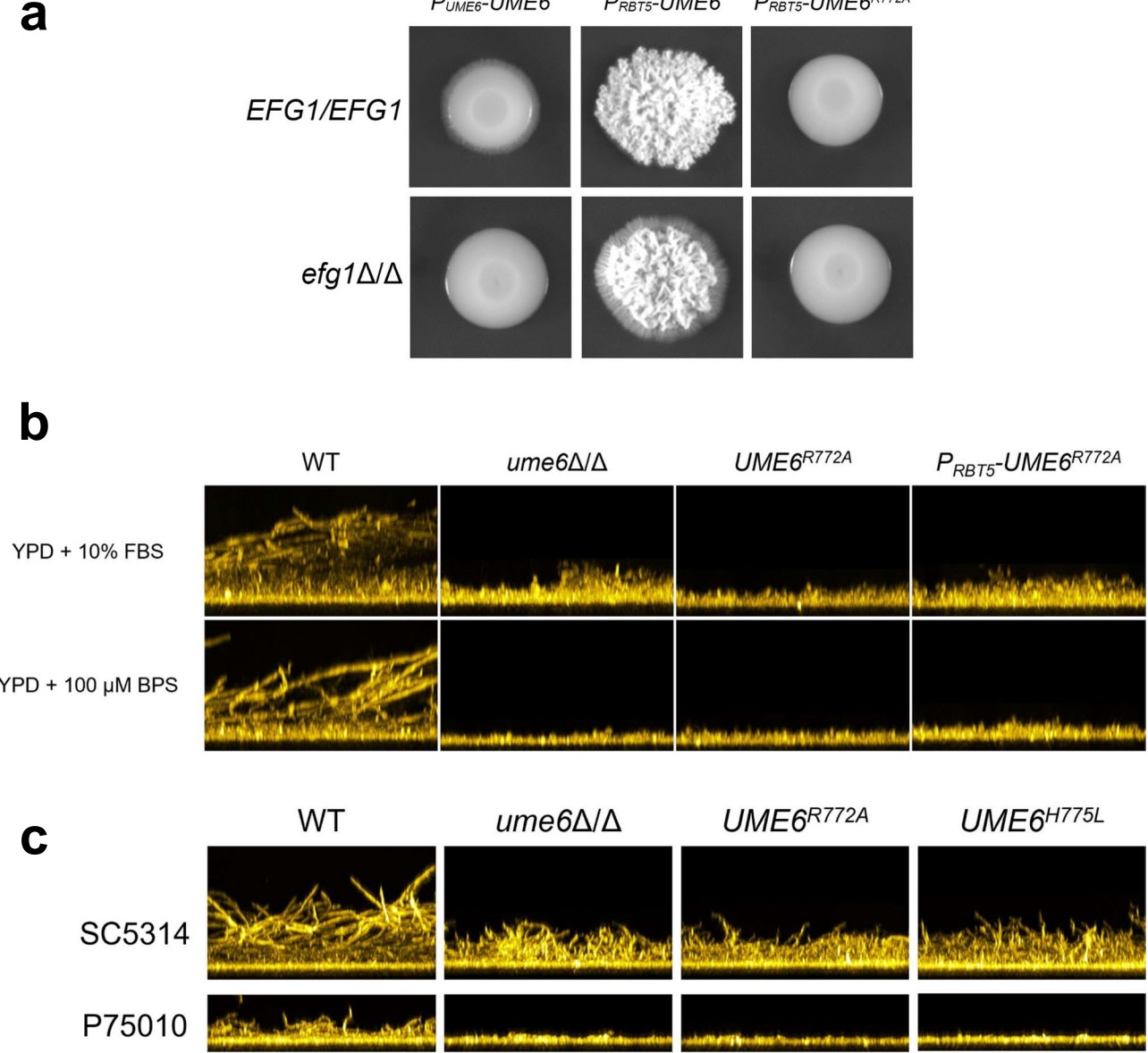

**Extended Data Fig. 3 | DNA-contact residues are essential for Ume6 function.** (a) Cells were grown on YPD plate at 37 °C for 3 days, then the plate was imaged. The images represent three independent experiments. (b) Biofilm formation under YPD + 10% FBS or YPD + 100 μM BPS for 24 h at 37 °C. Biofilms were stained with Calcofluor-White. The side-view projection images represent three independent experiments. (c) Biofilm formation under RPMI + 10% FBS medium for 24 h at 37 °C. Biofilms were stained with Calcofluor-White. The side-view projection images represent three independent experiments. All strains used in this figure are homozygous for alleles indicated. The WT genotype is *UME6-HA/UME6-HA*.

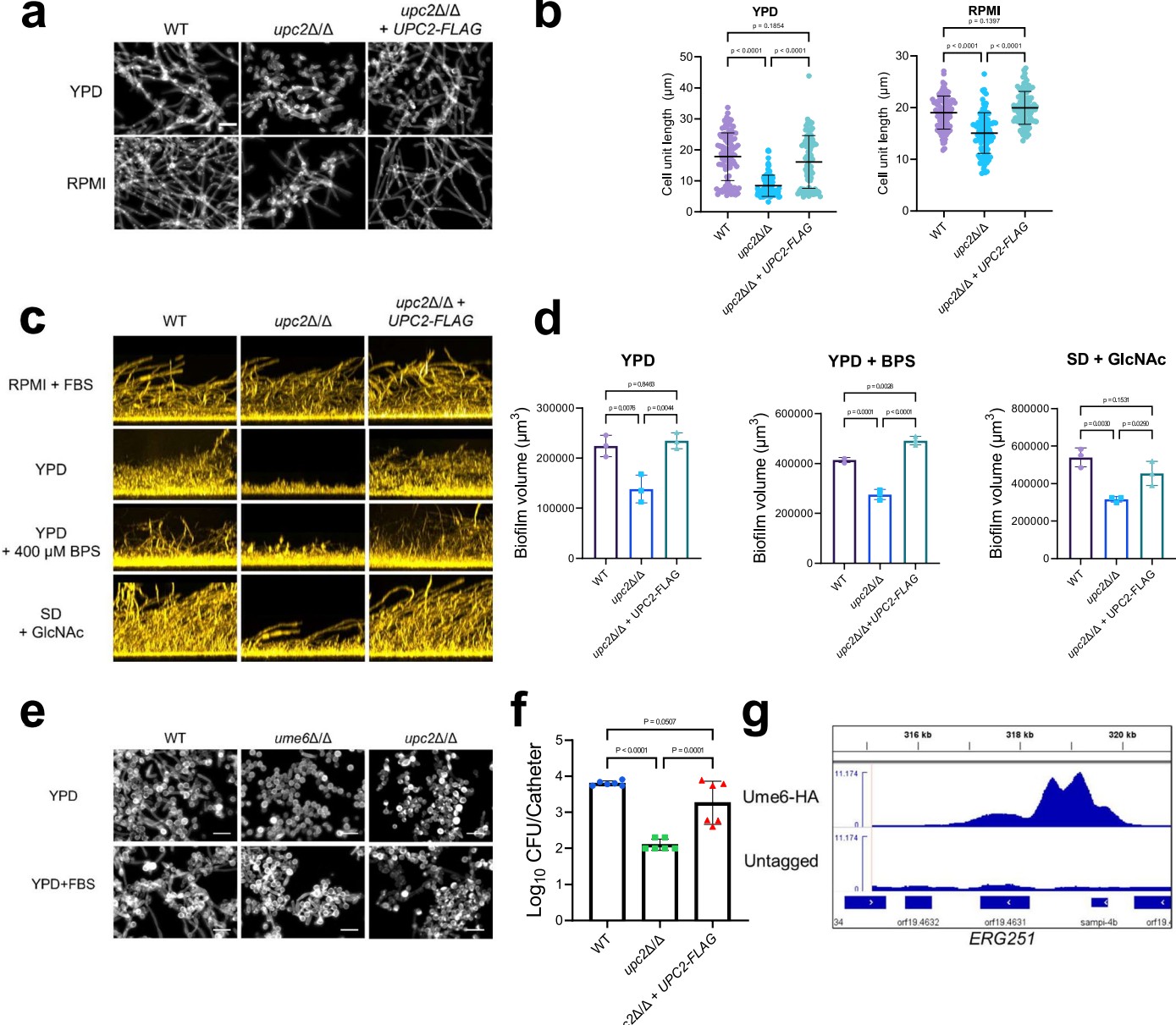

**Extended Data Fig. 4 | Upc2 promotes hypha and biofilm formation.**
**(a)** For filamentation assay, cells were grown in YPD at 37 °C for 4 h and stained with Calcofluor-White. White scale bars are 20 microns. The images represent three independent experiments. **(b)** The scatter plots indicate cell unit length of each strain. Cell lengths were measured using a minimum of 100 cells from 3 different fields. The images represent three independent experiments. Statistical significance was determined using one-way ANOVA (Tukey's multiple comparisons test). The line and error bars indicate the mean ± s.d. **(c)** For biofilm formation, cells were grown in indicated medium at 37 °C for 24 h, and biofilms were stained with Calcofluor-White. The side-view projection images represent three independent experiments. **(d)** Biofilm volume measurements. Statistical significance was determined using one-way ANOVA (Tukey's multiple comparisons test). Three biological replicates were used (n = 3). Values are

the mean ± s.d of three biological replicates. **(e)** For filamentation assay, cells were grown in YPD at 37 °C for 24 h under biofilm condition and stained with Calcofluor-White. White scale bars are 20 microns. The images represent three independent experiments. **(f)** SC5314 wild type, *upc2Δ/Δ* mutant, and complemented strain *upc2Δ/Δ + UPC2* were tested for *in vivo* biofilm formation in a rat venous catheter infection model. *C. albicans* cell counts per catheter were determined at 48 h post-infection. The graph presents six independent experiments. Statistical significance was determined using one-way ANOVA (Tukey's multiple comparisons test). Values are the mean ± s.d of three biological replicates. **(g)** IGV tracks indicate Ume6 binding sites from ChIP-seq data. The y-axis indicates read counts. The ChIP-seq tracks represent three biological replicates. All strains used in this figure are homozygous for alleles indicated. The WT genotype is *UME6-HA/UME6-HA*.

# Reporting Summary

## Statistics

For all statistical analyses, confirm that the following items are present in the figure legend, table legend, main text, or Methods section.

| n/a | Confirmed | |
|---|---|---|
| ☐ | ☒ | The exact sample size (*n*) for each experimental group/condition, given as a discrete number and unit of measurement |
| ☐ | ☒ | A statement on whether measurements were taken from distinct samples or whether the same sample was measured repeatedly |
| ☐ | ☒ | The statistical test(s) used AND whether they are one- or two-sided *Only common tests should be described solely by name; describe more complex techniques in the Methods section.* |
| ☒ | ☐ | A description of all covariates tested |
| ☒ | ☐ | A description of any assumptions or corrections, such as tests of normality and adjustment for multiple comparisons |
| ☐ | ☒ | A full description of the statistical parameters including central tendency (e.g. means) or other basic estimates (e.g. regression coefficient) AND variation (e.g. standard deviation) or associated estimates of uncertainty (e.g. confidence intervals) |
| ☐ | ☒ | For null hypothesis testing, the test statistic (e.g. *F*, *t*, *r*) with confidence intervals, effect sizes, degrees of freedom and *P* value noted *Give P values as exact values whenever suitable.* |
| ☒ | ☐ | For Bayesian analysis, information on the choice of priors and Markov chain Monte Carlo settings |
| ☒ | ☐ | For hierarchical and complex designs, identification of the appropriate level for tests and full reporting of outcomes |
| ☒ | ☐ | Estimates of effect sizes (e.g. Cohen's *d*, Pearson's *r*), indicating how they were calculated |

*Our web collection on statistics for biologists contains articles on many of the points above.*

## Software and code

Policy information about availability of computer code

| Data collection | Keyence BZ-X800 viewer was used for collection of microscopy data for biofilm assay. AxioVision software (version 4.8.2.0) was used for collection of microscopy data for filamentation assay. CFX Manager (version 3.1.1517.0823) was used for collection of both ChIP-qPCR and qRT-PCR data. |
|---|---|
| Data analysis | Details are included in the materials and methods section in the manuscript. Fiji/ImageJ (ver2.14.0) was used for visualization and measurement of biofilm volume and hyphal cell length. GraphPad Prism (ver9) Mapping: hisat2 (ver 2.0.5) Assembly: Stringtie (ver1.3.3b) Quantification: featureCounts (ver1.5.0-p3) Differential analysis: DESeq2 (ver1.22.1) VolcaNoseR (https://huygens.science.uva.nl/VolcaNoseR2/) IUPRED3 (https://iupred3.elte.hu/) |

For manuscripts utilizing custom algorithms or software that are central to the research but not yet described in published literature, software must be made available to editors and reviewers. We strongly encourage code deposition in a community repository (e.g. GitHub). See the Nature Portfolio guidelines for submitting code & software for further information.

## Data

Policy information about availability of data

All manuscripts must include a data availability statement. This statement should provide the following information, where applicable:

- Accession codes, unique identifiers, or web links for publicly available datasets
- A description of any restrictions on data availability
- For clinical datasets or third party data, please ensure that the statement adheres to our policy

Processed RNA-seq and ChIP-seq data are available in Table S1 and S2; raw data are available through NCBI SRA with accession numbers PRJNA1114072 (RNA-seq), PRJNA1114713 (Ume6-HA ChIP-seq), and PRJNA1114694 (Ume6-HA overexpression ChIP-seq).

## Research involving human participants, their data, or biological material

Policy information about studies with human participants or human data. See also policy information about sex, gender (identity/presentation), and sexual orientation and race, ethnicity and racism.

| | |
|---|---|
| Reporting on sex and gender | N/A |
| Reporting on race, ethnicity, or other socially relevant groupings | N/A |
| Population characteristics | N/A |
| Recruitment | N/A |
| Ethics oversight | N/A |

Note that full information on the approval of the study protocol must also be provided in the manuscript.

# Field-specific reporting

Please select the one below that is the best fit for your research. If you are not sure, read the appropriate sections before making your selection.

☒ Life sciences ☐ Behavioural & social sciences ☐ Ecological, evolutionary & environmental sciences

For a reference copy of the document with all sections, see nature.com/documents/nr-reporting-summary-flat.pdf

# Life sciences study design

All studies must disclose on these points even when the disclosure is negative.

| | |
|---|---|
| Sample size | No statistical methods were used to pre-determine sample sizes for any experiments in this study. For RNA-seq and ChIP-seq experiments, we employed three biological replicates. For biofilm assay, we employed three biological replicates. For ChIP-qPCR and qRT-PCR, we employed three biological replicates. For filamentation assay, we employed a hundred of hyphal cells from three independent images. For in vivo biofilm formation, we employed six biological replicates from two different experiments. These sample sizes are widely chosen for studies in our field. |
| Data exclusions | No data was excluded from analysis. |
| Replication | All experiments were subject to at least three biological replications unless otherwise stated. |
| Randomization | Randomization was not relevant for this study since samples were not allocated to experimental groups. |
| Blinding | Bilnding was not relevant for this study. |

# Reporting for specific materials, systems and methods

We require information from authors about some types of materials, experimental systems and methods used in many studies. Here, indicate whether each material, system or method listed is relevant to your study. If you are not sure if a list item applies to your research, read the appropriate section before selecting a response.

## Materials & experimental systems

| n/a | Involved in the study |
|-----|----------------------|
| ☐ | ☒ Antibodies |
| ☒ | ☐ Eukaryotic cell lines |
| ☒ | ☐ Palaeontology and archaeology |
| ☐ | ☒ Animals and other organisms |
| ☒ | ☐ Clinical data |
| ☒ | ☐ Dual use research of concern |
| ☒ | ☐ Plants |

## Methods

| n/a | Involved in the study |
|-----|----------------------|
| ☐ | ☒ ChIP-seq |
| ☒ | ☐ Flow cytometry |
| ☒ | ☐ MRI-based neuroimaging |

## Antibodies

| | |
|---|---|
| Antibodies used | Anti-HA (rabbit): polyclonal anti-HA tag antibody, abcam, catalog # ab9110, 7ul used for ChIP, 1ul used for Co-IP, 1:10000 dilution for Western blotting.<br>Anti-FLAG (mouse): monoclonal anti-FLAG M2 antibody, Sigma, catalog #F3165, 1:5000 dilution for Western blotting.<br>Goat anti-mouse: anti-Mouse IgG (H+L) HRP from goat, ThermoFisher, Catalog # 31430, 1:5000 dilution for Western blotting.<br>Goat anti-rabbit: anti-rabbit IgG-HRP from goat, Jackson ImmunoResearch, catalog # 111-035-144,  1:5000 dilution for Western blotting. |
| Validation | All anti-bodies used in this manuscript was validated in our previous publication.<br>Do E, Cravener MV, Huang MY, May G, McManus CJ, Mitchell AP. Collaboration between Antagonistic Cell Type Regulators Governs Natural Variation in the Candida albicans Biofilm and Hyphal Gene Expression Network. mBio. 2022 Oct 26;13(5):e0193722. doi: 10.1128/mbio.01937-22. Epub 2022 Aug 22. PMID: 35593746; PMCID: PMC9600859. |

## Animals and other research organisms

Policy information about studies involving animals; ARRIVE guidelines recommended for reporting animal research, and Sex and Gender in Research

| | |
|---|---|
| Laboratory animals | Sprague Dawley rats, 16-week old, 400 g |
| Wild animals | Wild animals were not used in this study. |
| Reporting on sex | Male mice were used for experiments. |
| Field-collected samples | The studies did not involve field collected samples. |
| Ethics oversight | All animal procedures were approved by the Institutional Animal Care and Use Committee at the University of Wisconsin-Madison according to the guidelines of the Animal Welfare Act, The Institute of Laboratory Animal Resources Guide for the care and Use of Laboratory Animals, and Public Health Service Policy. The approved animal protocol number is DA0031. |

Note that full information on the approval of the study protocol must also be provided in the manuscript.

## Plants

| | |
|---|---|
| Seed stocks | N/A |
| Novel plant genotypes | N/A |
| Authentication | N/A |

## ChIP-seq

### Data deposition

☒ Confirm that both raw and final processed data have been deposited in a public database such as GEO.

☒ Confirm that you have deposited or provided access to graph files (e.g. BED files) for the called peaks.

| Data access links | www.ncbi.nlm.nih.gov/bioproject/PRJNA1114713 |
| --- | --- |
| *May remain private before publication.* | www.ncbi.nlm.nih.gov/bioproject/PRJNA1114694 |
| | All processed data has been submitted in the supplemental data. |

| Files in database submission | ED1_R_S1_L001_R1_001.fastq.gz |
| --- | --- |
| | ED2_R_S2_L001_R1_001.fastq.gz |
| | ED3_R_S3_L001_R1_001.fastq.gz |
| | ED4_R_S4_L001_R1_001.fastq.gz |
| | ED5_R_S5_L001_R1_001.fastq.gz |
| | ED6_R_S6_L001_R1_001.fastq.gz |
| | ED7_R_S7_L001_R1_001.fastq.gz |
| | ED8_R_S8_L001_R1_001.fastq.gz |
| | ED9_R_S9_L001_R1_001.fastq.gz |
| | ED10_R_S10_L001_R1_001.fastq.gz |
| | ED11_R_S11_L001_R1_001.fastq.gz |
| | ED12_R_S12_L001_R1_001.fastq.gz |
| | ED13_R_S13_L001_R1_001.fastq.gz |
| | ED14_R_S14_L001_R1_001.fastq.gz |
| | ED15_R_S15_L001_R1_001.fastq.gz |
| | ED16_R_S16_L001_R1_001.fastq.gz |
| | ED17_R_S17_L001_R1_001.fastq.gz |
| | ED18_R_S18_L001_R1_001.fastq.gz |
| | ED19_R_S19_L001_R1_001.fastq.gz |
| | ED20_R_S20_L001_R1_001.fastq.gz |
| | ED21_R_S21_L001_R1_001.fastq.gz |
| | ED22_R_S22_L001_R1_001.fastq.gz |
| | ED1_R_S1_L001_R2_001.fastq.gz |
| | ED2_R_S2_L001_R2_001.fastq.gz |
| | ED3_R_S3_L001_R2_001.fastq.gz |
| | ED4_R_S4_L001_R2_001.fastq.gz |
| | ED5_R_S5_L001_R2_001.fastq.gz |
| | ED6_R_S6_L001_R2_001.fastq.gz |
| | ED7_R_S7_L001_R2_001.fastq.gz |
| | ED8_R_S8_L001_R2_001.fastq.gz |
| | ED9_R_S9_L001_R2_001.fastq.gz |
| | ED10_R_S10_L001_R2_001.fastq.gz |
| | ED11_R_S11_L001_R2_001.fastq.gz |
| | ED12_R_S12_L001_R2_001.fastq.gz |
| | ED13_R_S13_L001_R2_001.fastq.gz |
| | ED14_R_S14_L001_R2_001.fastq.gz |
| | ED15_R_S15_L001_R2_001.fastq.gz |
| | ED16_R_S16_L001_R2_001.fastq.gz |
| | ED17_R_S17_L001_R2_001.fastq.gz |
| | ED18_R_S18_L001_R2_001.fastq.gz |
| | ED19_R_S19_L001_R2_001.fastq.gz |
| | ED20_R_S20_L001_R2_001.fastq.gz |
| | ED21_R_S21_L001_R2_001.fastq.gz |
| | ED22_R_S22_L001_R2_001.fastq.gz |
| | ED1_4349_S1_L001_R1_001.fastq.gz |
| | ED2_4349_S1_L001_R1_001.fastq.gz |
| | ED3_4349_S2_L001_R1_001.fastq.gz |
| | ED4_4349_S2_L001_R1_001.fastq.gz |
| | ED5_4349_S3_L001_R1_001.fastq.gz |
| | ED6_4349_S3_L001_R1_001.fastq.gz |
| | ED7_4349_S4_L001_R1_001.fastq.gz |
| | ED8_4349_S4_L001_R1_001.fastq.gz |
| | ED9_4349_S5_L001_R1_001.fastq.gz |
| | ED16_4349_S8_L001_R1_001.fastq.gz |
| | ED17_4349_S9_L001_R1_001.fastq.gz |
| | ED18_4349_S9_L001_R1_001.fastq.gz |
| | ED19_4349_S10_L001_R1_001.fastq.gz |
| | ED20_4349_S10_L001_R1_001.fastq.gz |
| | ED1_4349_S1_L001_R2_001.fastq.gz |
| | ED2_4349_S1_L001_R2_001.fastq.gz |
| | ED3_4349_S2_L001_R2_001.fastq.gz |
| | ED4_4349_S2_L001_R2_001.fastq.gz |
| | ED5_4349_S3_L001_R2_001.fastq.gz |
| | ED6_4349_S3_L001_R2_001.fastq.gz |
| | ED7_4349_S4_L001_R2_001.fastq.gz |
| | ED8_4349_S4_L001_R2_001.fastq.gz |
| | ED9_4349_S5_L001_R2_001.fastq.gz |
| | ED16_4349_S8_L001_R2_001.fastq.gz |
| | ED17_4349_S9_L001_R2_001.fastq.gz |
| | ED18_4349_S9_L001_R2_001.fastq.gz |

ED19_4349_S10_L001_R2_001.fastq.gz
ED20_4349_S10_L001_R2_001.fastq.gz

Genome browser session
(e.g. UCSC)

Not applicable

# Methodology

Replicates

All ChIP-seq experiments was performed with biological triplicate samples.

Sequencing depth

ChIP-seq libraries were sequenced on Illumina NextSeq 2000 sequencing system (Illumina, 2 x 51 nt, paired-end).
Raw read-depth ranged from 24 M to 65 M read pairs per sample. To normalize read depth, we downsampled the number of aligned reads to 5 M pairs (10 M total reads) for each sample before calling peaks.

Antibodies

Anti-HA rabbit polyclonal (abcam: ab9110) was used for ChIP-seq experiments.

Peak calling parameters

MACS2 (v 2.1.0.20151222; default parameters.)

Data quality

Threshold of False discovery rate (FDR) was set at 0.05 and the peaks consistent across the replicates were further used in the annotation.

Software

bowtie2 (v 2.1.0)
samtools (v1.10)
MACS2 (v 2.1.0.20151222)
mspc (v 5.4.0)
DiffBind (Galaxy Version 2.10.0+galaxy0)
HOMER (v 4.11)
Integrative Genomics Viewer v2.11.0

