## [Peer Review File · Nature Microbiology]

Ume6 protein complexes connect morphogenesis, adherence, and hypoxic genes to shape *Candida albicans* biofilm architecture

Corresponding Author: Dr Aaron Mitchell

Version 0:

Reviewer comments:

Reviewer #1

(Remarks to the Author)

Review Nature Microbiology Do et al

Diverse binding partners drive Ume6 function in *Candida albicans* biofilm formation

This interesting and exciting manuscript reports findings that illuminate our understanding of how biofilms form and function in the pathogenic fungus *Candida albicans*. This species is one of the three most common systemic human fungal pathogens, a WHO critical priority group species, and a significant cause of morbidity and mortality globally. Its ability to form biofilms is integral to both its biology and its pathogenesis, and formation of biofilms contributes to antimicrobial drug resistance. Considerable effort has elucidated both the genetic network that enables biofilm formation as well as the regulatory circuitry that governs this developmental process. The studies presented here significantly extend this understanding by revealing the key and central mechanistic role for the Ume6 zinc finger transcription factor, whose precise role had been somewhat enigmatic prior to this study. The study is well done, rigorous, the manuscript a pleasure to read from start to finish. The study is the result of a long-term collaboration between two leading luminaries in this field, one of whom is an active infectious disease physician scientist providing key clinical expertise as a further foundation for these efforts.

The work is based on careful and detailed RNA-seq, ChIP-seq, protein-protein interaction analysis, as well as mutational and domain analysis. The findings reveal that Ume6 is central to the biofilm regulatory cascade and contributes to control expression of genes that are major biofilm determinants and hyphal associated factors, as well as the biofilm master regulators including Bcr1, Efg1, Flo8, Tec1, and others. An interesting and surprising finding is that Ume6 is brought to promoters that have been pre-determined by a series of other biofilm transcriptional regulators including Efg1, Ndt80, and Upc2. Ume6 was found to form complexes with these three factors in co-immunoprecipitation analyses. Quite interestingly, the prion like domain of Ume6 was not required to form these protein-protein complexes, suggesting that the Ume6-DNA-transcription factor complexes differ from complexes formed by Efg1 and other biofilm regulators that involve formation of nuclear condensates. Moreover, because Ume6 interacts also with Upc2, a key hypoxic response regulator, the model is advanced that Ume6 serves as a bridge between maintenance of hyphal and biofilm formation and cellular physiological responses within the biofilm to the low oxygen environment.

The authors make a persuasive argument that facets of Ume6 function revealed here will likely be conserved in other *Candida* species, such as *Candida auris*, further increasing the impact of these findings. The conceptual advance of revealing how a factor with relatively low binding affinity for DNA operates in the context of other licensing factors, essentially hitchhiking to its destinations, is of broad and general relevance to understanding control of gene expression in any system or organism. With some minor revisions, it is this reviewer's opinion and recommendation that this outstanding study should be published in Nature Microbiology.

Specific comments

1. Lines 173/174 From this reviewer's perspective, "pull down" is jargon that might well be replaced with: "capture", "recover", "precipitate" or some other more scientific term.
2. Line 174, a word may be missing: "detected presence" might be changed to "detected the presence".
3. Line 193, the words create and created might be better reserved for religious connotations. How about "We engineered or we produced or we constructed or we generated"?

Reviewer #2

(Remarks to the Author)

This is a nicely written manuscript that addresses a fundamental question pertaining to a central regulator of *Candida albicans* morphogenesis and biofilm formation: Ume6. Ume6 is a zinc cluster transcription factor (TF) that has been historically the subject of extensive research and here the authors address a central feature of its transcriptional regulatory activity: 1) the direct targets that Ume6 regulates and 2) the identity of the binding/functionally-interacting partners. Consequently, this manuscript is foundational in terms of the mechanistic underpinnings of Ume6 function and - from a broader view - brings fundamental novelties to our understanding of how TFs form complexes to regulate gene expression.

Specific comments:

-Line 79: It could be better here to include reference 17 and introduce in a couple of words the RBT5 promoter as a tool for regulated expression (Mao et al. 2022) and better define the rationale for using an overexpression approach in an *efg1*^{-/-} strain.

-Figure 1A: It is striking that Efg1 controls almost all basal expression levels of UME6. Was this previously reported in the literature? If not, this should be highlighted. However, Ume6-HA is still able to bind DNA (albeit at very low levels) in the absence of Efg1 (Figure 2B), somehow contrasting with the undetectability of the Ume6-HA fusion by Western Blot analyses in *efg1*^{-/-} mutant. This should be explained.

-I believe the prediction of Ume6 DNA-binding motif could be further refined by performing the analysis with different motif discovery algorithms, one powerful algorithm is PeakMotifs (https://rsat.france-bioinformatique.fr/fungi/peak-motifs_form.cgi). How do results look like here with the set of 400 peaks centered on summits?

-The authors strikingly show that Ume6 mediates interaction with Efg1, Ndt80 and Upc2 through a region encompassing amino acids 720-843 (C-terminal part) and does not involve, for example, a predicted coiled-coil domain located at the N-terminus of Ume6 (amino acids 61-81). This is really an exciting finding. The C-terminal region carries the DNA-binding domain (zinc cluster motif in positions 763-807) as well as a region of low compositional complexity, from aa 718 to 749 (32 amino acids), predicted to be part of a larger disordered region (aa 580 to 759). It would be nice to further discuss the properties of this 32-amino acid sequence and how it could potentially mediate the TF-TF interactions. Site-directed mutagenesis/deletion analyses in this tiny fragment could probably further delineate/resolve the TF-TF interaction motifs.

Minor comments/typos:

-Line 129: "dependence on".

-Line 386: "in detail".

Reviewer #3

(Remarks to the Author)

The manuscript submitted by Do and co-workers describe how diverse binding partners drive the function of the transcription factor Ume6 in *Candida albicans* biofilm formation. They have used state of the art methods including RNA-seq, ChiP-seq, Co-IP to show that Ume6 binds to the promoters of hyphae- and biofilm-associated genes and activates them, e.g. during growth in RPMI-FBS medium. They also show that the binding motif of Ume6 is similar to the binding motifs of biofilm master regulators such as Efg1 and Ndt80. Further analysis revealed that Ume6 can physically interact with Efg1, Ndt80 and also Upc2 and that these three transcription factors are involved in the binding of Ume6 to target promoters. Finally, the authors speculate in both, abstract and discussion, that Ume6 might somehow connect the gene expression during the early phases of biofilm formation (hyphal growth, adherence) and later phases, especially hypoxic response genes. This conclusion is mainly drawn from the shown interaction between Ume6 and Upc2 as the latter is not only known as a regulator of genes involved in ergosterol biosynthesis but also of hypoxic response genes.

The transcription factor Ume6 is one of the most analyzed transcription factors in the field of *Candida albicans* and its role in the complex regulatory network that controls the expression of genes involved in hyphal morphogenesis, adherence and biofilm formation is well characterized. However, there are many open questions, especially in terms of physical interaction with other proteins and promoters of target genes. Here, the authors provide convincing data of how Ume6 binds to target regions under hyphae- inducing conditions, how the binding motif looks like and which consequences this could have for possible interactions. Indeed, the authors were able to prove that Ume6 is part of a network with other transcription factors with a lot of possibilities how they interact with each other and how they regulate each other. While the proof of physical interaction with Efg1 and Ndt80 is new and important, the finding of the interaction with Upc2 is more surprising and interesting for further research as this observation could lead to an involvement of Ume6 in ergosterol biosynthesis and hypoxic response.

Although I really liked the manuscript, there are several points which might be considered prior to a publication:

1) The experiments are technically sound and I agree with most of the conclusions drawn from the results. However, I think that

the Material and Methods section is very short and does not allow a complete understanding of the performed experiments. I understand that there are space restrictions, but the authors should add some crucial information, e.g. condition and time points of RNA isolation as well as amount of RNA used for RT-qPCR.

2) From the title on, biofilm is an important topic in this manuscript. However, after reading the Material and Methods section I realized that RNA isolation, chromatin extraction and protein isolation were performed with shaking cultures and not from biofilms directly. Indeed, the authors neither wrote in the main part that this was done directly with biofilms, I think they should make this a bit clearer that all the conclusions in terms of RNA-seq, ChIP-seq, Co-IP and Western Blot are drawn from shaking cultures and not biofilms. Is there a reason why biofilms were not used for these steps? At least RNA extraction from biofilms was described several years ago, e.g. by Nobile et al., 2012 in Cell (doi: 10.1016/j.cell.2011.10.048). I still think that the conclusions are true and reliable but it should be addressed that only the microscopy data are from biofilms directly. If I got it completely wrong, I apologize.

3) The authors used the *C. albicans* RBT5 promoter to drive UME6 overexpression instead of the Tet on/ off systems previously used by other groups. Could they exclude that binding by hyphae-associated regulators which control the expression of RBT5 gene indirectly affects the expression rates of UME6 in their setting?

4) The authors themselves describe that expression of genes like ALS3 and ECE1 in the *efg1* deletion mutant overexpressing UME6 was lower than in wild type overexpressing UME6 (Figure 1c). My question is, if the authors are sure that ALS3 and ECE1 are expressed at functional relevant levels? Is there really Als3 protein on the hyphal cell membrane and is there really a secretion of candidalysin? I ask because it was shown that although both genes were upregulated in the *ahr1* and *tup1* deletion mutants, in both cases neither Als3 protein on the surface nor candidalysin secretion was detected which indicated that a certain threshold of expression must be surpassed (Ruben et al., 2020 mBio, DOI: 10.1128/mBio.00206-20).

5) As already mentioned, the authors claim that Ume6 might connect early stages of biofilm formation with later ones, especially the hypoxic response genes required for cell growth under low oxygen concentrations. It is mentioned twice (abstract, discussion) and I am not sure if they can really show enough data to make this conclusion. It all depends on the shown interaction of Ume6 with Upc2. But if it would be the case that this interaction indeed has an effect on hypoxic response genes, then I would expect that the authors already see this in their data. Why don't they show it? Or if this is not the case then they should at least write that an involvement of Ume6 in the regulation of hypoxic response genes via Upc2-binding is possible but not proven yet. Maybe this also demands another experimental setting and direct isolation of RNA from late stage biofilms.

Decision Letter:

3rd March 2025

Dear Aaron,

Thank you for your patience while your manuscript "Diverse binding partners drive Ume6 function in *Candida albicans* biofilm formation" was under peer-review at Nature Microbiology, and while I discussed your revision plan with our chief editor. Your manuscript has been seen by 3 referees, whose expertise and comments you will find at the end of this email. Although they find your work of some potential interest, they have raised a number of concerns that will need to be addressed before we can consider publication of the work in Nature Microbiology.

In particular, while referees #1 and #2 are more positive, referee #3 raises several concerns, including technical concerns. Specifically, referee #3 has concern that "RNA isolation, chromatin extraction and protein isolation were performed with shaking cultures and not from biofilms directly". I have now discussed your revision plan and the proposed experiments (RNA-seq, Co-IP, and ChIP-qPCR under biofilm conditions) with our chief editor, and we feel this is a promising approach to address the referees' concerns.

Therefore, should further experimental data allow you to address these criticisms, we would be happy to look at a revised manuscript.

Please include a data availability statement as a separate section after Methods but before references, under the heading "Data Availability". This section should inform readers about the availability of the data used to support the conclusions of your study. This information includes accession codes to public repositories (data banks for protein, DNA or RNA sequences, microarray, proteomics data etc...), references to source data published alongside the paper, unique identifiers such as URLs to data repository entries, or data set DOIs, and any other statement about data availability. At a minimum, you should include the

following statement: "The data that support the findings of this study are available from the corresponding author upon request", mentioning any restrictions on availability. If DOIs are provided, we also strongly encourage including these in the Reference list (authors, title, publisher (repository name), identifier, year). For more guidance on how to write this section please see: <http://www.nature.com/authors/policies/data/data-availability-statements-data-citations.pdf>

* If you have not done so already we suggest that you begin to revise your manuscript so that it conforms to our Article format instructions at <http://www.nature.com/nmicrobiol/info/final-submission>. Refer also to any guidelines provided in this letter.

When submitting the revised version of your manuscript, please pay close attention to our [href="https://www.nature.com/nature-portfolio/editorial-policies/image-integrity">Digital Image Integrity Guidelines. and to the following points below:](https://www.nature.com/nature-portfolio/editorial-policies/image-integrity)

EXTENDED DATA FIGURES

Link Redacted

Note: This url links to your confidential homepage and associated information about manuscripts you may have submitted or be reviewing for us. If you wish to forward this e-mail to co-authors, please delete this link to your homepage first.

Nature Microbiology is committed to improving transparency in authorship. As part of our efforts in this direction, we are now requesting that all authors identified as 'corresponding author' on published papers create and link their Open Researcher and Contributor Identifier (ORCID) with their account on the Manuscript Tracking System (MTS), prior to acceptance. This applies to primary research papers only. ORCID helps the scientific community achieve unambiguous attribution of all scholarly contributions. You can create and link your ORCID from the home page of the MTS by clicking on 'Modify my Springer Nature account'. For more information please visit www.springernature.com/orcid.

If you wish to submit a suitably revised manuscript we would hope to receive it within 6 months. If you cannot send it within this time, please let us know.

Yours sincerely,

Reviewer Expertise:

Referee #1: Fungal pathogens, fungal genetics

Referee #2: ChIP-seq

Referee #3: Medical mycology, transcriptional regulation, Ume6, biofilms

Reviewer Comments:

Reviewer #1 (Remarks to the Author):

Review Nature Microbiology Do et al

Diverse binding partners drive Ume6 function in *Candida albicans* biofilm formation

This interesting and exciting manuscript reports findings that illuminate our understanding of how biofilms form and function in the pathogenic fungus *Candida albicans*. This species is one of the three most common systemic human fungal pathogens, a WHO critical priority group species, and a significant cause of morbidity and mortality globally. Its ability to form biofilms is integral to both its biology and its pathogenesis, and formation of biofilms contributes to antimicrobial drug resistance. Considerable effort has elucidated both the genetic network that enables biofilm formation as well as the regulatory circuitry that governs this developmental process. The studies presented here significantly extend this understanding by revealing the key and central mechanistic role for the Ume6 zinc finger transcription factor, whose precise role had been somewhat enigmatic prior to this study. The study is well done, rigorous, the manuscript a pleasure to read from start to finish. The study is the result of a long-term collaboration between two leading luminaries in this field, one of whom is an active infectious disease physician scientist providing key clinical expertise as a further foundation for these efforts.

The work is based on careful and detailed RNA-seq, ChIP-seq, protein-protein interaction analysis, as well as mutational and domain analysis. The findings reveal that Ume6 is central to the biofilm regulatory cascade and contributes to control expression of genes that are major biofilm determinants and hyphal associated factors, as well as the biofilm master regulators including Bcr1, Efg1, Flo8, Tec1, and others. An interesting and surprising finding is that Ume6 is brought to promoters that have been pre-determined by a series of other biofilm transcriptional regulators including Efg1, Ndt80, and Upc2. Ume6 was found to form complexes with these three factors in co-immunoprecipitation analyses. Quite interestingly, the prion like domain of Ume6 was not required to form these protein-protein complexes, suggesting that the Ume6-DNA-transcription factor complexes differ from complexes formed by Efg1 and other biofilm regulators that involve formation of nuclear condensates. Moreover, because Ume6 interacts also with Upc2, a key hypoxic response regulator, the model is advanced that Ume6 serves as a bridge between maintenance of hyphal and biofilm formation and cellular physiological responses within the biofilm to the low oxygen environment.

The authors make a persuasive argument that facets of Ume6 function revealed here will likely be conserved in other *Candida* species, such as *Candida auris*, further increasing the impact of these findings. The conceptual advance of revealing how a factor with relatively low binding affinity for DNA operates in the context of other licensing factors, essentially hitchhiking to its destinations, is of broad and general relevance to understanding control of gene expression in any system or organism. With some minor revisions, it is this reviewer's opinion and recommendation that this outstanding study should be published in *Nature Microbiology*.

Specific comments

1. Lines 173/174 From this reviewer's perspective, "pull down" is jargon that might well be replaced with: "capture", "recover", "precipitate" or some other more scientific term.
2. Line 174, a word may be missing: "detected presence" might be changed to "detected the presence".
3. Line 193, the words create and created might be better reserved for religious connotations. How about "We engineered or we produced or we constructed or we generated"?

Reviewer #2 (Remarks to the Author):

This is a nicely written manuscript that addresses a fundamental question pertaining to a central regulator of *Candida albicans* morphogenesis and biofilm formation: Ume6. Ume6 is a zinc cluster transcription factor (TF) that has been historically the subject of extensive research and here the authors address a central feature of its transcriptional regulatory activity: 1) the direct targets that Ume6 regulates and 2) the identity of the binding/functionally-interacting partners. Consequently, this manuscript is foundational in terms of the mechanistic underpinnings of Ume6 function and - from a broader view - brings fundamental novelties to our understanding of how TFs form complexes to regulate gene expression.

Specific comments:

-Line 79: It could be better here to include reference 17 and introduce in a couple of words the RBT5 promoter as a tool for regulated expression (Mao et al. 2022) and better define the rationale for using an overexpression approach in an *efg1*^{-/-} strain.

-Figure 1A: It is striking that Efg1 controls almost all basal expression levels of UME6. Was this previously reported in the literature? If not, this should be highlighted. However, Ume6-HA is still able to bind DNA (albeit at very low levels) in the absence of Efg1 (Figure 2B), somehow contrasting with the undetectability of the Ume6-HA fusion by Western Blot analyses in *efg1*^{-/-} mutant. This should be explained.

-I believe the prediction of Ume6 DNA-binding motif could be further refined by performing the analysis with different motif discovery algorithms, one powerful algorithm is PeakMotifs (https://rsat.france-bioinformatique.fr/fungi/peak-motifs_form.cgi). How do results look like here with the set of 400 peaks centered on summits?

-The authors strikingly show that Ume6 mediates interaction with Efg1, Ndt80 and Upc2 through a region encompassing amino acids 720-843 (C-terminal part) and does not involve, for example, a predicted coiled-coil domain located at the N-terminus of Ume6 (amino acids 61-81). This is really an exciting finding. The C-terminal region carries the DNA-binding domain (zinc cluster motif in positions 763-807) as well as a region of low compositional complexity, from aa 718 to 749 (32 amino acids), predicted to be part of a larger disordered region (aa 580 to 759). It would be nice to further discuss the properties of this 32-amino acid sequence and how it could potentially mediate the TF-TF interactions. Site-directed mutagenesis/deletion analyses in this tiny fragment could probably further delineate/resolve the TF-TF interaction motifs.

Minor comments/typos:

-Line 129: "dependence on".

-Line 386: "in detail".

Reviewer #3 (Remarks to the Author):

The manuscript submitted by Do and co-workers describe how diverse binding partners drive the function of the transcription factor Ume6 in *Candida albicans* biofilm formation. They have used state of the art methods including RNA-seq, ChIP-seq, Co-IP to show that Ume6 binds to the promoters of hyphae- and biofilm-associated genes and activates them, e.g. during growth in RPMI-FBS medium. They also show that the binding motif of Ume6 is similar to the binding motifs of biofilm master regulators such as Efg1 and Ndt80. Further analysis revealed that Ume6 can physically interact with Efg1, Ndt80 and also Upc2 and that these three transcription factors are involved in the binding of Ume6 to target promoters. Finally, the authors speculate in both, abstract and discussion, that Ume6 might somehow connect the gene expression during the early phases of biofilm formation (hyphal growth, adherence) and later phases, especially hypoxic response genes. This conclusion is mainly drawn from the shown interaction between Ume6 and Upc2 as the latter is not only known as a regulator of genes involved in ergosterol biosynthesis but also of hypoxic response genes.

The transcription factor Ume6 is one of the most analyzed transcription factors in the field of *Candida albicans* and its role in the complex regulatory network that controls the expression of genes involved in hyphal morphogenesis, adherence and biofilm formation is well characterized. However, there are many open questions, especially in terms of physical interaction with other proteins and promoters of target genes. Here, the authors provide convincing data of how Ume6 binds to target regions under hyphae- inducing conditions, how the binding motif looks like and which consequences this could have for possible interactions. Indeed, the authors were able to prove that Ume6 is part of a network with other transcription factors with a lot of possibilities how they interact with each other and how they regulate each other. While the proof of physical interaction with Efg1 and Ndt80 is new and important, the finding of the interaction with Upc2 is more surprising and interesting for further research as this observation could lead to an involvement of Ume6 in ergosterol biosynthesis and hypoxic response.

Although I really liked the manuscript, there are several points which might be considered prior to a publication:

1) The experiments are technically sound and I agree with most of the conclusions drawn from the results. However, I think that the Material and Methods section is very short and does not allow a complete understanding of the performed experiments. I understand that there are space restrictions, but the authors should add some crucial information, e.g. condition and time points of RNA isolation as well as amount of RNA used for RT-qPCR.

2) From the title on, biofilm is an important topic in this manuscript. However, after reading the Material and Methods section I realized that RNA isolation, chromatin extraction and protein isolation were performed with shaking cultures and not from biofilms directly. Indeed, the authors neither wrote in the main part that this was done directly with biofilms, I think they should make this a bit clearer that all the conclusions in terms of RNA-seq, ChIP-seq, Co-IP and Western Blot are drawn from shaking cultures and not biofilms. Is there a reason why biofilms were not used for these steps? At least RNA extraction from biofilms was described several years ago, e.g. by Nobile et al., 2012 in *Cell* (doi: 10.1016/j.cell.2011.10.048). I still think that the conclusions are true and reliable but it should be addressed that only the microscopy data are from biofilms directly. If I got it completely wrong, I apologize.

3) The authors used the *C. albicans* RBT5 promoter to drive UME6 overexpression instead of the Tet on/ off systems previously used by other groups. Could they exclude that binding by hyphae-associated regulators which control the expression of RBT5 gene indirectly affects the expression rates of UME6 in their setting?

4) The authors themselves describe that expression of genes like ALS3 and ECE1 in the *efg1* deletion mutant overexpressing UME6 was lower than in wild type overexpressing UME6 (Figure 1c). My question is, if the authors are sure that ALS3 and ECE1 are expressed at functional relevant levels? Is there really Als3 protein on the hyphal cell membrane and is there really a secretion of candidalysin? I ask because it was shown that although both genes were upregulated in the *ahr1* and *tup1* deletion mutants, in both cases neither Als3 protein on the surface nor candidalysin secretion was detected which indicated that a certain threshold of expression must be surpassed (Ruben et al., 2020 *mBio*, DOI: 10.1128/mBio.00206-20).

5) As already mentioned, the authors claim that Ume6 might connect early stages of biofilm formation with later ones, especially the hypoxic response genes required for cell growth under low oxygen concentrations. It is mentioned twice (abstract, discussion) and I am not sure if they can really show enough data to make this conclusion. It all depends on the shown interaction of Ume6 with Upc2. But if it would be the case that this interaction indeed has an effect on hypoxic response genes, then I would expect that the authors already see this in their data. Why don't they show it? Or if this is not the case then they

should at least write that an involvement of Um6 in the regulation of hypoxic response genes via Upc2-binding is possible but not proven yet. Maybe this also demands another experimental setting and direct isolation of RNA from late stage biofilms.

Version 1:

Reviewer comments:

Reviewer #2

(Remarks to the Author)

The authors addressed all comments raised by this reviewer.

Reviewer #3

(Remarks to the Author)

I want to thank the authors to put so much additional efforts into the study to address my concerns. I recommend the acceptance of the revised manuscript.

Decision Letter:

Our ref: NMICROBIOL-24113526A

6th June 2025

Dear Aaron,

Thank you for submitting your revised manuscript "Diverse binding partners drive Ume6 function in *Candida albicans* biofilm formation" (NMICROBIOL-24113526A). It has now been seen by the original referees and their comments are below. The reviewers find that the paper has improved in revision, and therefore we'll be happy in principle to publish it in Nature Microbiology, pending minor revisions to comply with our editorial and formatting guidelines.

We are now performing detailed checks on your paper and will send you a checklist detailing our editorial and formatting requirements in about two weeks. Please do not upload the final materials and make any revisions until you receive this additional information from us.

Thank you again very much for your interest in Nature Microbiology. Please do not hesitate to contact me if you have any questions.

Sincerely,

Reviewer #2 (Remarks to the Author):

The authors addressed all comments raised by this reviewer.

Reviewer #3 (Remarks to the Author):

I want to thank the authors to put so much additional efforts into the study to address my concerns. I recommend the acceptance of the revised manuscript.

Version 2:

Decision Letter:

26th June 2025

Dear Aaron,

I am pleased to accept your Article "Ume6 protein complexes connect morphogenesis, adherence, and hypoxic genes to shape *Candida albicans* biofilm architecture" for publication in Nature Microbiology. Thank you for having chosen to submit your work to us and many congratulations.

Authors may need to take specific actions to achieve <https://www.springernature.com/gp/open-research/funding/policy-compliance-faqs> compliance with funder and institutional open access mandates. If your research is supported by a funder that requires immediate open access (e.g. according to <https://www.springernature.com/gp/open-research/plan-s-compliance>) Plan S principles then you should select the gold OA route, and we will direct you to the compliant route where possible. For authors selecting the subscription publication route, the journal's standard licensing terms will need to be accepted, including <https://www.nature.com/nature-portfolio/editorial-policies/self-archiving-and-license-to-publish> self-archiving policies. Those licensing terms will supersede any other terms that the author or any third party may assert apply to any version of the manuscript.

Congratulations once again and I look forward to seeing the article published.

With kind regards,

P.S. Click on the following link if you would like to recommend Nature Microbiology to your librarian
<http://www.nature.com/subscriptions/recommend.html#forms>

** Visit the Springer Nature Editorial and Publishing website at www.springernature.com/editorial-and-publishing-jobs for more information about our career opportunities. If you have any questions please click here.**

Reviewer #1

Review Nature Microbiology Do et al

Diverse binding partners drive Ume6 function in *Candida albicans* biofilm formation

This interesting and exciting manuscript reports findings that illuminate our understanding of how biofilms form and function in the pathogenic fungus *Candida albicans*. This species is one of the three most common systemic human fungal pathogens, a WHO critical priority group species, and a significant cause of morbidity and mortality globally. Its ability to form biofilms is integral to both its biology and its pathogenesis, and formation of biofilms contributes to antimicrobial drug resistance. Considerable effort has elucidated both the genetic network that enables biofilm formation as well as the regulatory circuitry that governs this developmental process. The studies presented here significantly extend this understanding by revealing the key and central mechanistic role for the Ume6 zinc finger transcription factor, whose precise role had been somewhat enigmatic prior to this study. The study is well done, rigorous, the manuscript a pleasure to read from start to finish. The study is the result of a long-term collaboration between two leading luminaries in this field, one of whom is an active infectious disease physician scientist providing key clinical expertise as a further foundation for these efforts.

We thank you for expressing such great enthusiasm for our study!

The work is based on careful and detailed RNA-seq, ChIP-seq, protein-protein interaction analysis, as well as mutational and domain analysis. The findings reveal that Ume6 is central to the biofilm regulatory cascade and contributes to control expression of genes that are major biofilm determinants and hyphal associated factors, as well as the biofilm master regulators including Bcr1, Efg1, Flo8, Tec1, and others. An interesting and surprising finding is that Ume6 is brought to promoters that have been pre-determined by a series of other biofilm transcriptional regulators including Efg1, Ndt80, and Upc2. Ume6 was found to form complexes with these three factors in co-immunoprecipitation analyses. Quite interestingly, the prion like domain of Ume6 was not required to form these protein-protein complexes, suggesting that the Ume6-DNA-transcription factor complexes differ from complexes formed by Efg1 and other biofilm regulators that involve formation of nuclear condensates. Moreover, because Ume6 interacts also with Upc2, a key hypoxic response regulator, the model is advanced that Ume6 serves as a bridge between maintenance of hyphal and biofilm formation and cellular physiological responses within the biofilm to the low oxygen environment.

We are grateful for this concise summary.

The authors make a persuasive argument that facets of Ume6 function revealed here will likely be conserved in other *Candida* species, such as *Candida auris*, further increasing the impact of these findings. The conceptual advance of revealing how a factor with relatively low binding affinity for DNA operates in the context of other licensing factors, essentially hitchhiking to its destinations, is of broad and general relevance to understanding control of gene expression in any system or organism.

With some minor revisions, it is this reviewer's opinion and recommendation that this outstanding study should be published in Nature Microbiology.

Thank you again for your enthusiasm!

Specific comments

1. Lines 173/174 From this reviewer's perspective, "pull down" is jargon that might well be replaced with: "capture", "recover", "precipitate" or some other more scientific term.

We modified the text as follows: "Anti-HA antibody was used to precipitate tagged Ume6-HA, and anti-Flag antibody detected presence of each Flag-tagged..."

2. Line 174, a word may be missing: "detected presence" might be changed to "detected the presence".

We now say: "...detected the presence..."

3. Line 193, the words create and created might be better reserved for religious connotations. How about "We engineered or we produced or we constructed or we generated"?

We now say: "...we constructed mutant allele..."

Reviewer #2

This is a nicely written manuscript that addresses a fundamental question pertaining to a central regulator of *Candida albicans* morphogenesis and biofilm formation: Ume6. Ume6 is a zinc cluster transcription factor (TF) that has been historically the subject of extensive research and here the authors address a central feature of its transcriptional regulatory activity: 1) the direct targets that Ume6 regulates and 2) the identity of the binding/functionally-interacting partners. Consequently, this manuscript is foundational in terms of the mechanistic underpinnings of Ume6 function and - from a broader view - brings fundamental novelties to our understanding of how TFs form complexes to regulate gene expression.

Thank you for this highly supportive appraisal!

Specific comments:

-Line 79: It could be better here to include reference 17 and introduce in a couple of words the *RBT5* promoter as a tool for regulated expression (Mao et al. 2022) and better define the rationale for using an overexpression approach in an *efg1*^{-/-} strain.

We have added this statement to the paragraph: "The *RBT5* promoter, which is active in RPMI + FBS medium, was used for overexpression¹⁷." We prefer not to discuss the construct further because it risks diverting attention from the main focus.

-Figure 1A: It is striking that Efg1 controls almost all basal expression levels of UME6. Was this previously reported in the literature? If not, this should be highlighted.

However, Ume6-HA is still able to bind DNA (albeit at very low levels) in the absence of

Efg1 (Figure 2B), somehow contrasting with the undetectability of the Ume6-HA fusion by Western Blot analyses in *efg1*^{-/-} mutant. This should be explained.

Severely reduced *UME6* RNA levels in *efg1Δ/Δ* mutants was shown previously by RNA-seq analysis of multiple strains (Huang et al. 2019 and Cravener et al., 2023). In addition, our current RNA-seq analysis also showed a significant reduction of *UME6* RNA levels in the *efg1Δ/Δ* mutant compared to WT (-4.8 Log₂FC). We have now modified the third sentence of Results to say: “The *efg1Δ/Δ* mutant is defective in filamentation, biofilm formation, and expression of hypha-associated genes, including *UME6*¹⁶.”

Yes, we were surprised that we could detect Ume6-DNA binding in an *efg1Δ/Δ* mutant as well! The low background with the ChIP anti-HA antibody allows quite sensitive detection of DNA signals. We note that average FPKM values for *UME6* in WT and *efg1Δ/Δ* mutant are 162.33 and 5.66, respectively, suggesting that there is a basal level of Ume6 expression in the *efg1Δ/Δ* mutant.

-I believe the prediction of Ume6 DNA-binding motif could be further refined by performing the analysis with different motif discovery algorithms, one powerful algorithm is PeakMotifs (https://rsat.france-bioinformatique.fr/fungi/peak-motifs_form.cgi). How do results look like here with the set of 400 peaks centered on summits?

We have tested the motif analysis using PeakMotifs, and it showed peaks for motifs corresponding to Ndt80, Efg1, and Upc2, as we found with HOMER. It also detected a motif for Bcr1, which makes perfect sense since Bcr1 and Efg1 can form a phase-separated complex (Ganser et al., PMID: 38091321). The Rim101 motif may be functionally meaningful as well, though the E-value is weaker than the others. We prefer to explore a Rim101 connection in the future, since its function is not closely tied to biofilm formation.

Motifs	E-value	Known TF
 o1gpo_6at_mkx3_m1 417 sites	7.8e-09	Ndt80
 o1gpo_6at_mkx3_m2 235 sites	4.3e-06	Bcr1
 o1gpo_6at_mkx3_m3 382 sites	4.6e-06	Efg1
 o1gpo_6at_mkx3_m4 530 sites	5.9e-06	Upc2
 o1gpo_6at_mkx3_m5 356 sites	1.2e-05	Rim101

-The authors strikingly show that Ume6 mediates interaction with Efg1, Ndt80 and Upc2 through a region encompassing amino acids 720-843 (C-terminal part) and does not involve, for example, a predicted coiled-coil domain located at the N-terminus of Ume6 (amino acids 61-81). This is really an exciting finding. The C-terminal region carries the DNA-binding domain (zinc cluster motif in positions 763-807) as well as a region of low compositional complexity, from aa 718 to 749 (32 amino acids), predicted to be part of a larger disordered region (aa 580 to 759). It would be nice to further discuss the properties of this 32-amino acid sequence and how it could potentially mediate the TF-TF interactions. Site-directed mutagenesis/deletion analyses in this tiny fragment could probably further delineate/resolve the TF-TF interaction motifs.

We agree that this region is quite interesting! In response to this comment we constructed a strain expressing Ume6⁷⁵⁹⁻⁸⁴³-HA, a derivative that harbors only C-terminal DNA binding domain and lacks the 32 amino acid stretch. We tested it in a Co-IP assay, in which cells were grown in RPMI + FBS at 37C for 4 hours. The results below showed that Ume6⁷⁵⁹⁻⁸⁴³-HA can physically interact with Ndt80, but not with other TFs, Efg1 and Upc2. However, the Ume6⁷⁵⁹⁻⁸⁴³-HA protein level is considerably lower than Ume6-HA, indicating that the shortened Ume6 protein may possess an unstable structure or be degraded. For this reason we are not highly confident of the negative results regarding interaction with Efg1 and Upc2.

In addition, the sequence of that 32 amino acid region is not well conserved among *Candida* species. Our suspicion is that the DNA binding domain is the region responsible for all three protein-protein interactions, as seems to be the case with Ndt80. Analysis will require site-directed mutagenesis, a more time-consuming approach.

Because we are skeptical that the 32 amino acid region mediates protein-protein interactions, we prefer not to discuss its properties in the manuscript.

Minor comments/typos:

-Line 129: "dependence on".
Fixed – thank you!

-Line 386: "in detail".
Fixed – thank you!

Reviewer #3

The manuscript submitted by Do and co-workers describe how diverse binding partners drive the function of the transcription factor Ume6 in *Candida albicans* biofilm formation. They have used state of the art methods including RNA-seq, ChiP-seq, Co-IP to show that Ume6 binds to the promoters of hyphae- and biofilm-associated genes and activates them, e.g. during growth in RPMI-FBS medium. They also show that the binding motif of Ume6 is similar to the binding motifs of biofilm master regulators such as Efg1 and Ndt80. Further analysis revealed that Ume6 can physically interact with Efg1, Ndt80 and also Upc2 and that these three transcription factors are involved in the binding of Ume6 to target promoters. Finally, the authors speculate in both, abstract and discussion, that Ume6 might somehow connect the gene expression during the early phases of biofilm formation (hyphal growth, adherence) and later phases, especially hypoxic response genes. This conclusion is mainly drawn from the shown interaction between Ume6 and Upc2 as the latter is not only known as a regulator of genes involved in ergosterol biosynthesis but also of hypoxic response genes.

The transcription factor Ume6 is one of the most analyzed transcription factors in the field of *Candida albicans* and its role in the complex regulatory network that controls the expression of genes involved in hyphal morphogenesis, adherence and biofilm formation is well characterized. However, there are many open questions, especially in terms of physical interaction with other proteins and promoters of target genes. Here, the authors provide convincing data of how Ume6 binds to target regions under hyphae-inducing conditions, how the binding motif looks like and which consequences this could have for possible interactions. Indeed, the authors were able to prove that Ume6 is part of a network with other transcription factors with a lot of possibilities how they interact with each other and how they regulate each other. While the proof of physical interaction with Efg1 and Ndt80 is new and important, the finding of the interaction with Upc2 is more surprising and interesting for further research as this observation could lead to an involvement of Ume6 in ergosterol biosynthesis and hypoxic response.
Thank you for this exceptionally clear summary!

Although I really liked the manuscript, there are several points which might be considered prior to a publication:

1) The experiments are technically sound and I agree with most of the conclusions drawn from the results. However, I think that the Material and Methods section is very short and does not allow a complete understanding of the performed experiments. I understand that there are space restrictions, but the authors should add some crucial information, e.g. condition and time points of RNA isolation as well as amount of RNA used for RT-qPCR.

Apologies! We have added detail including time points to the Methods sections entitled RNA extraction and sequencing, qRT-PCR, Western blot assay, and Co-immunoprecipitation assay.

2) From the title on, biofilm is an important topic in this manuscript. However, after reading the Material and Methods section I realized that RNA isolation, chromatin extraction and protein isolation were performed with shaking cultures and not from biofilms directly. Indeed, the authors neither wrote in the main part that this was done directly with biofilms, I think they should make this a bit clearer that all the conclusions in terms of RNA-seq, ChIP-seq, Co-IP and Western Blot are drawn from shaking cultures and not biofilms. Is there a reason why biofilms were not used for these steps? At least RNA extraction from biofilms was described several years ago, e.g. by Nobile et al., 2012 in Cell (doi: 10.1016/j.cell.2011.10.048). I still think that the conclusions are true and reliable but it should be addressed that only the microscopy data are from biofilms directly.

If I got it completely wrong, I apologize.

You are correct that we used shaking cultures for the analyses in our initial submission. In response to this excellent point though, we have added Co-IP and gene expression data obtained with biofilm protein extracts (Fig. 3d and 6g,h in the revised manuscript, and shown here for convenience). Cells were grown in RPMI + FBS under biofilm condition at 37°C for 24 hours. The Co-IP result (representing three independent experiments) indicates that Ume6 interacts with Efg1, Ndt80 and Upc2 under biofilm condition. Interaction with Efg1 seems slightly weaker under biofilm conditions than under planktonic conditions (see the revised manuscript figure please).

The outcome is reasonable biologically because Efg1 is required early in biofilm formation in order to make hyphae and activate numerous adhesin genes and biofilm-associated genes, including *UME6*. On the other hand, Ndt80 and Upc2 are not required for hypha formation under the strong hypha-inducing conditions we used in this study.

We also examined expression of several hypha-associated genes, as well as diverse *ERG* and hypoxia-induced genes described by Sellam et al. (PMID: 24681685) and Burgain et al. (PMID: 32102943). In this case we grew biofilms at 37°C for 24 hours in YPD medium, a condition in which the *upc2Δ/Δ* mutant presents a defective phenotype. RNA levels were measured by QRT-PCR. Among 14 genes that require Upc2 for full levels of expression, we found 12 genes that also depended upon Ume6. The impact of the *upc2Δ/Δ* mutation was always as great or greater than that of the *ume6Δ/Δ* mutation, which is consistent with the idea that Ume6 augments Upc2-dependent gene activation. Overall, our results argue that Ume6 is indeed required for full expression of Upc2-dependent genes under biofilm conditions.

3) The authors used the *C. albicans* RBT5 promoter to drive UME6 overexpression instead of the Tet on/ off systems previously used by other groups. Could they exclude that binding by hyphae-associated regulators which control the expression of RBT5 gene indirectly affects the expression rates of UME6 in their setting?

We tested Ume6-HA levels by Western blot analysis, and Ume6-HA DNA-binding activity by ChIP-qPCR, in cells in YPD + 400uM BPS at 37°C for 4 hours. These are the same conditions used in Figure 5, chosen because this is the medium in which a *upc2Δ/Δ* mutant has a clear biofilm defect (Figure 5a). Western blot analysis showed that RBT5 promoter-driven Ume6-HA protein levels are similar in each mutant strain:

For ChIP-qPCR assays, we examined three Ume6-bound promoters: *CDR1*, *VPH2*, and *MPT5*. Binding of Ume6-HA to all three promoters was comparable across the mutant strain panel:

4) The authors themselves describe that expression of genes like ALS3 and ECE1 in the *efg1* deletion mutant overexpressing UME6 was lower than in wild type overexpressing UME6 (Figure 1c). My question is, if the authors are sure that ALS3 and ECE1 are expressed at functional relevant levels? Is there really Als3 protein on the hyphal cell membrane and is there really a secretion of candidalysin? I ask because it was shown that although both genes were upregulated in the *ahr1* and *tup1* deletion mutants, in both cases neither Als3 protein on the surface nor candidalysin secretion was detected which indicated that a certain threshold of expression must be surpassed (Ruben et al., 2020 mBio, DOI: 10.1128/mBio.00206-20).

This is an excellent point: we do not have direct information about the levels of any of the target gene products under Ume6 control. We did not pursue this question in the current study because we focused on the transcriptional regulators themselves, yielding results that all three reviewers found exciting and novel. We agree that a complete picture of functional impact will require examination of target gene product levels, modifications, and functional activity.

5) As already mentioned, the authors claim that Ume6 might connect early stages of biofilm formation with later ones, especially the hypoxic response genes required for cell growth under low oxygen concentrations. It is mentioned twice (abstract, discussion) and I am not sure if they can really show enough data to make this conclusion. It all depends on the shown interaction of Ume6 with Upc2. But if it would be the case that this interaction indeed has an effect on hypoxic response genes, then I would expect that the authors already see this in their data. Why don't they show it? Or if this is not the case then they should at least write that an involvement of Ume6 in the regulation of hypoxic response genes via Upc2-binding is possible but not proven yet. Maybe this also demands another experimental setting and direct isolation of RNA from late stage biofilms.

We believe that this question, whether Ume6 controls hypoxic genes, is addressed in the response to point 2 above. In brief, we used QRT-PCR assays of biofilm cells, and found that Ume6 is required for full levels of expression of 12 genes that are under Upc2 control, among 14 genes assayed. We have included a portion of the figure here, showing hypoxic response genes specifically, for convenience.